# Observation error analysis for the WIVERN W-band Doppler conically scanning spaceborne radar via end-to-end simulations

Alessandro Battaglia[1,2,3], Paolo Martire[1], Eric Caubet[4], Laurent Phalippou[4], Fabrizio Stesina[1], Pavlos Kollias[5,6], and Anthony Illingworth[7]

[1]Politecnico of Torino, Italy
[2]University of Leicester, UK
[3]National Centre for Earth Observation, UK
[4]Thales Alenia Space, Toulouse, France
[5]Department of Environmental and Climate Sciences, Brookhaven National Laboratory, Upton, NY, USA
[6]Division of Atmospheric Sciences, Stony Brook University, Stony Brook, NY, USA
[7]University of Reading, Reading, UK

**Correspondence:** Alessandro Battaglia
alessandro_battaglia@polito.it

**Abstract.** The WIVERN (WInd VElocity Radar Nephoscope) mission, now in Phase-0 of the ESA Earth Explorer program, promises to complement Doppler wind lidar by globally observing, for the first time, vertical profiles of winds in cloudy areas. This work describes an initial assessment of the performances of the WIVERN conically scanning 94 GHz Doppler radar, the only payload of the mission. The analysis is based on an end-to-end simulator characterised by novel features tailored to the WIVERN radar: the conically scanning geometry; the inclusion of cross-polarization effects and of the simulation of a radiometric mode; the applicability to global cloud model outputs via an orbital model; the incorporation of a mispointing model accounting for thermo-elastic distortions, microvibrations, star-trackers uncertainties, etc.; the inclusion of the surface clutter. Some of the simulator capabilities are showcased for a case study involving a full rotational scan of the instrument.

Preliminary findings show that mispointing errors associated with the antenna azimuthal mispointing are expected to be lower than 0.3 ms$^{-1}$ (and strongly dependent on the antenna azimuthal scanning angle), wind shear and non-uniform beam filling errors have generally negligible biases when full antenna revolutions are considered, non-uniform beam filling causes random errors strongly dependent on the antenna azimuthal scanning angle but typically lower than 1 ms$^{-1}$ and cross-talk effects are well predictable so that areas affected by strong cross-talk noise can be flagged. Overall the quality of the Doppler velocities appears to strongly depend on several factors: the strength of the cloud reflectivity, the antenna pointing direction relative to the satellite motion, the presence of strong reflectivity and/or wind gradients, the strength of the surface clutter. The end-to-end simulations suggest that total wind errors meet the mission requirements in a good portion of the clouds detected by the WIVERN radar,

The simulator will be used for studying trade-offs for the different WIVERN configurations under consideration during Phase 0 (e.g. different antenna sizes, pulse lengths, antenna patterns, . . . ). Thanks to its modular structure the simulator can be easily adapted to different orbits, different scanning geometries and different frequencies.

# 1   Introduction

Accurate forecasts save lives, support emergency management and the mitigation of impacts, thus preventing losses from severe weather while creating substantial revenue (Bauer et al., 2015). Windstorms are the largest contributor to economic losses caused by weather related hazards, resulting in approximately 500 billion USD of global damage over the last decade (https://www.ncdc.noaa.gov/billions/). Together with floods they are the costliest natural hazards in Europe: they account for more than 30% (60%) of total (insured) losses (https://ec.europa.eu/jrc/sites/jrcsh/files/pesetaiv_task_13_windstorms_final_report.pdf). The Aeolus wind lidar has demonstrated a large impact in reducing forecast errors when assimilated by European Weather Forecasting Centers (Rennie et al., 2021). In addition to winds, cloud and precipitation measurements remain key for both Numerical Weather Prediction (NWP) applications and for advancing understanding of cloud processes and their role in climate simulations.

The WIVERN (WInd VElocity Radar Nephoscope) concept has been recently proposed within the ESA Earth Explorer 11 call in order to strengthen the wind, cloud and precipitation observation capability of the Global Observing System. The mission has been selected for Phase 0 studies. It hinges upon a single instrument: a dual-polarization Doppler W-band scanning cloud radar with a 3 m circular aperture non-deployable main reflector. The WIVERN antenna conically scans around nadir at an off-nadir angle of 38° at 12 revolution per minute (rpm). This rotation speed implies the use of one horn for transmission and another one for reception. Flying on a 500-km orbit, the instrument provides a swath of 800 km (see Fig. 1).

The aim of the mission is to complement Doppler lidar winds acquired in clear sky conditions and from the tops of optically thick clouds (Rennie et al., 2021) and other wind observations (profiles by radio soundings, at cloud top via geostationary observation derived atmospheric motion vectors, close to the ocean surface by scatterometers) by observations in areas of optically thick clouds, critical for cyclogenesis, that cannot be seen by optical sensors. Observations in these areas have the largest potential to improve forecasts (McNally, 2002). Therefore the WIVERN mission is expected to provide (Illingworth et al., 2018b, 2020):

- unprecedented wind observations inside tropical cyclones and mid-latitude windstorms that will routinely reveal the dynamic structure of such destructive systems;

- observations of convective motions that will validate the representation of convection in models;

- global profiles of cloud properties and precipitation over an 800 km swath that will better quantify the hydrological cycle and the atmospheric and surface energy budget;

- first direct observation of tropospheric winds that will underpin the predictions of transport and dispersion of trace gases and pollutants in atmospheric chemistry and air quality models.

These advances in the observational capabilities are expected to address three science objectives (Illingworth et al., 2018b, 2020).

1. To extend the lead time of useful prediction skills of hazardous weather (e.g., wind-storm, cyclones, floods) by direct assimilation of wide-swath winds from clouds and profiles of radar reflectivity of clouds and precipitation into numerical weather prediction (NWP) models.

2. To improve numerical models by providing new metrics and observational verification to assess different NWP parameterisation schemes within such models. NWP and climate models use similar schemes so better NWP models will also augment confidence in climate models.

3. To establish a benchmark for the climate record of cloud profiles, global solid/light precipitation and, for the first time, in-cloud winds, crucial for a better quantification of the Earth's hydrological cycle, and energy budgets, with a significant reduction in sampling errors of current and planned cloud radar missions.

**Table 1.** WMO (World Meteorological Organisation) requirements for horizontal winds for numerical weather prediction (NWP) and the expected performance of WIVERN.

|  | Uncertainty | Horizontal Resolution | Vertical Resolution | Observing Cycle |
|---|---|---|---|---|
| Goal | $2 \text{ ms}^{-1}$ | 15 km | 0.5 km | 1 h |
| Breakthrough | $3 \text{ ms}^{-1}$ | 100 km | 1 km | 6 h |
| Threshold | $5 \text{ ms}^{-1}$ | 100 km | 3 km | 12 h |
| WIVERN | $2 \text{ ms}^{-1}$ | 20 km | 0.64 km | 1 day$^\star$ |

$^\star$ Global average between $\pm 82°$ latitude.

WMO requirements for data assimilation into global NWP (Illingworth et al., 2018a) can be found at OSCAR (https://www.wmo-sat.info/oscar/) and are summarised in Tab. 1. The threshold of 12 h for the observing cycle is quite demanding; three scatterometers with 1200-km swaths can approach this revisit time. Noticeably, the Aeolus non-scanning narrow swath clear sky wind measurements are having a significant effect despite their typical clear-sky uncertainty of 4-5 $\text{ms}^{-1}$ and their coarse sampling (Rennie et al., 2021). Thus, even winds with uncertainty above the WMO threshold and with sampling below threshold have proved extremely valuable for NWP. Horanyi et al. (2014) showed that assimilating winds biased by 1-2 $\text{ms}^{-1}$ when the random error is around 2 $\text{ms}^{-1}$ would degrade the forecast so a bias of less than 1 $\text{ms}^{-1}$ should be added to the specifications of Tab. 1.

In order to achieve these targets WIVERN will adopt:

1. polarization diversity (i.e. the use of successive pulses with independent H and V polarization, Pazmany et al. (1999)) in order to overcome both the range-Doppler dilemma and the short decorrelation times produced by the Doppler fading associated with the low Earth orbiting satellite velocity (Battaglia et al., 2013);

2. a large antenna (3 m) in order to achieve a narrow beam, thus a fine vertical resolution and fewer issues related to non-uniform beam filling (NUBF) biases (Tanelli et al., 2002).

Previous studies (Illingworth et al., 2018b; Battaglia et al., 2018), based on the CloudSat climatology of cloud reflectivities, have demonstrated that the WIVERN radar should provide 1-2 million wind observations per day that satisfy the WMO "goal" of 2 ms$^{-1}$ precision. However it is important to define a rigorous framework where to assess the accuracy and precision of Doppler velocities. For instance errors introduced by satellite mispointing induced by orbital-dependent thermo-elastic distortion of the antenna, by the solar array drive mechanism microvibrations, by the rotating antenna vibration, etc., can seriously affect space-borne Doppler velocity measurements, as previously studied in Doppler scanning radars (Ardhuin et al., 2019) and in Doppler lidars (Weiler et al., 2021). Furthermore Battaglia et al. (2018) used 2D slant path profiles reconstructed from CloudSat and therefore did not implement the 3D scanning geometry of the WIVERN satellite. A full 3D framework is required to evaluate the importance of non-uniform beam filling errors and to assess how the quality of the Doppler velocity signal will depend on the antenna scanning viewing angle.

End-to-end (E2E) simulators are paramount tools for evaluating instrument performances in preparatory mission studies. They provide a high-fidelity performance prediction of the overall system. The focus of this study is in the mission performance assessment and error budget computation with a detailed partitioning of the different error contributors. Several radar simulators have been developed in the recent years to simulate space-borne atmospheric radars (e.g. Haynes et al. (2007); Matsui et al. (2013); Dellaripa et al. (2021)), including Doppler capabilities (e.g. Kollias et al. (2014); Sy et al. (2014)) as envisaged for the EarthCARE W-band Doppler radar (Illingworth et al., 2015). Doppler velocity estimates for that system will be based on the pulse pair technique (Doviak and Zrnić, 2006). The novelty of this work is that our radar simulator is tailored to conically scanning Doppler radars adopting polarization diversity. If selected, the WIVERN radar will be the first radar in space to ever adopt such technology. Therefore radar simulators have not yet included such novel features. The simulator also incorporates a model accounting for mispointing as potentially caused by different sources like thermo-elastic distortions, micro-vibrations, star-trackers uncertainties. Finally, it includes an orbital model with the possibility of changing orbit and thus viewing geometry. Sect. 2 provides a detailed description of all the modules of the E2E simulator whereas Sect. 3 presents some applications, with examples extracted from a case study and a first assessment of some of the errors related to the Doppler velocity measurements. Conclusions and future work are discussed in Sect. 4.

## 2  The E2E simulator

Our simulator capitalizes on recent refinements of radar simulators developed within different ESA projects. In particular it benefits from the inclusion of polarization diversity pulse pair processing and wide swath scanning (Battaglia et al., 2013; Battaglia and Kollias, 2015), of the effect of the cross-talk (Wolde et al., 2019) between the H and V channels caused by strongly reflective depolarising targets (e.g. the melting layer or the surface clutter) and the simulation of passive mode to provide brightness temperatures at W-band (Battaglia and Panegrossi, 2020). A simplified 2D-version of the simulator has recently been applied to CloudSat observations and co-located ECMWF 3D winds to provide an intial assessment of errors introduced by different sources related to aliasing, averaging and to the noise in the estimators of the Doppler spectra moments (Battaglia et al., 2018).

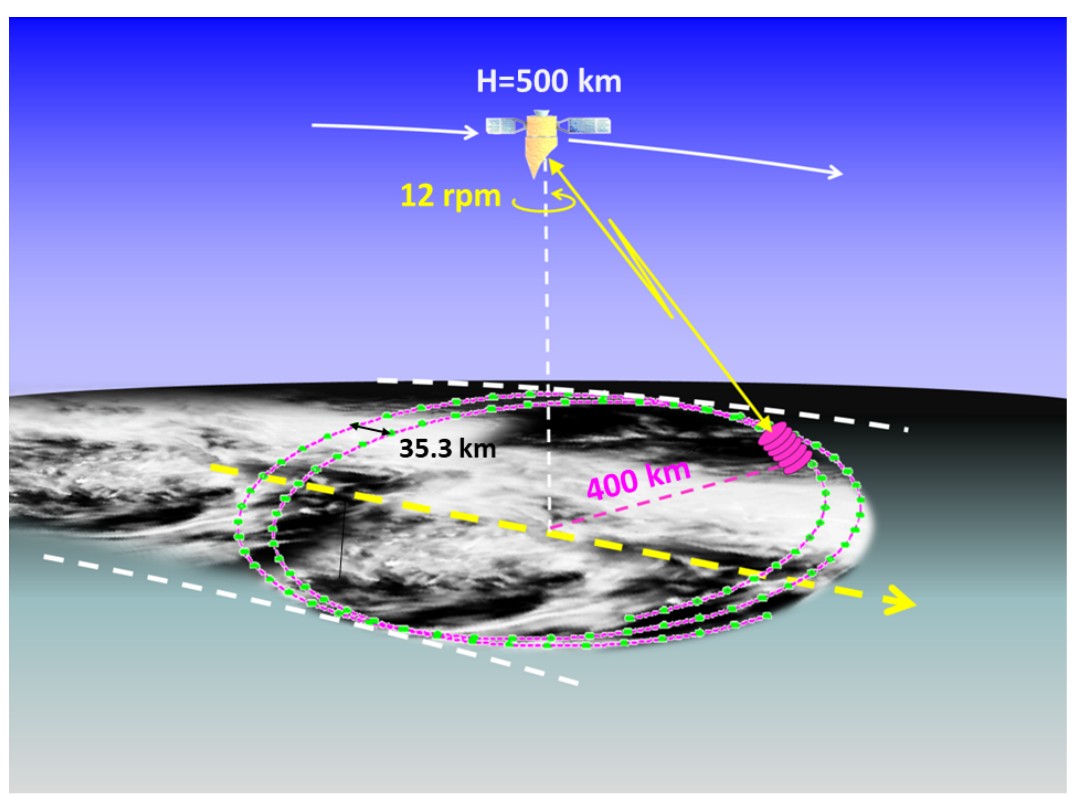

**Figure 1.** Artistic impression of the WIVERN concept: a 94-GHz Doppler radar with 3-m antenna scanning at 12 rpm tracing out a cycloidal track with an incidence angle of $41.6°$.

The simulator developed in this work can cope with data produced by state-of-the-art high-resolution cloud-resolving models as the basis for creating scenes that are used as input to the various instrument simulation modules. These outputs can be linked with sun-synchronous orbits produced by an orbital model derived from the two-body problem theory, with the addition of $J_2$ orbital perturbations. The user can modify the initial date and duration of orbital propagation and the orbital parameters. This provides the ability to simulate the satellite overpasses and the measurements for the given viewing geometry. In this control environment, forward and retrieval models can be evaluated and compared against the "truth" of the input model scene. Similarly, each error source can be evaluated separately based on the assumption that, as a first approximation, the different error sources can be assumed independent, so that the total quadratic error (bias) can be computed as a quadratic sum (an absolute sum) of the different errors (Battaglia and Kollias, 2015). For instance, the satellite motion NUBF-induced errors can be estimated by computing the velocities running the simulator with or without satellite motion and then taking the differences of the two (Battaglia et al., 2018).

A schematic for the overall structure of the simulator is depicted in Fig. 2 with a list of current and potential additional capabilities tabulated in Tab. 2. A global model provides high resolution 3D scenes with clouds and winds; outputs of the global model are used as inputs of a forward model that computes ideal profiles of W-band co- and cross-polar reflectivities and Doppler velocities. Note that the forward model is based on the single-scattering assumption. Multiple scattering effects are known to play an important role both for the reflectivity and the Doppler velocity signal in deep convective regions in presence of high attenuation (Battaglia et al., 2010b; Battaglia and Tanelli, 2011). The forward model outputs are then combined in a pulse-pair signal processing module which adds the proper noise levels to produce WIVERN outputs (H and V-channel reflectivities and line of sight Doppler velocities). Our tool simulates mean quantities and their errors as computed from well established radar theory (Doviak and Zrnić, 2006) for the specific polarization diversity pulse pair processing (Pazmany et al., 1999). These estimates have been been validated by an airborne field campaign (Wolde et al., 2019). Other simulators that compute $I$ and $Q$ time series (Battaglia et al., 2013; Kollias et al., 2014) are avoided here because of their high computational time.

The description of the different modules of the simulator is detailed in the following subsections. The radar specifics used throughout this paper are the ones recently proposed to the ESA Earth Explorer 11 and are listed in Tab. 3.

**Table 2.** Current and future capabilities of the WIVERN E2E simulator.

| Capability | Current | Future/Desirable |
|---|---|---|
| Model input | Global (4.3 km hor. res.) | Global (<1 km hor. res.)[†] |
| Surface backscattering model | Constant over ocean/land | Linked to surface properties (roughness, vegetation type, soil moisture, etc) |
| Simulated radar variables[*] | $Z_{co}, v_D, LDR$ | $Z_{DR}, A_{DP}, K_{DP}, \rho_{hv}$ |
| Multiple scattering | None | Based on Hogan and Battaglia (2008) |

[†] Currently such models are not available and represent a challenge for memory and/or computation time requirements. [*] The meaning of these variables is discussed in the text.

## 2.1 The System for Atmospheric Modeling (SAM) global storm resolving model

The Global Storm-Resolving Models (GSRM) (Stevens et al., 2019; Satoh et al., 2019) are a new class of high resolution global numerical models that explicitly simulate small scales of motions coupled to large-scale circulation systems. This allows GSRMs to explicitly resolve deep convection and thus overcome challenges arising from deep convection parameterizations (Kendon et al., 2017). The first intercomparison of GSRMs was conducted in the context of the DYAMOND (the DYnamics of the Atmospheric general circulation On Non-hydrostatic Domains) project (Stevens et al., 2019).

Here, output from a GSRM that participated in the DYAMOND project, the System for Atmospheric Modeling (SAM, Khairoutdinov and Randall (2003) which employs an anelastic form of the non-hydrostatic equations was used as input to the WIVERN radar simulator. The SAM has a horizontal resolution of 4.3 km and 74 vertical layers. Details of the SAM model

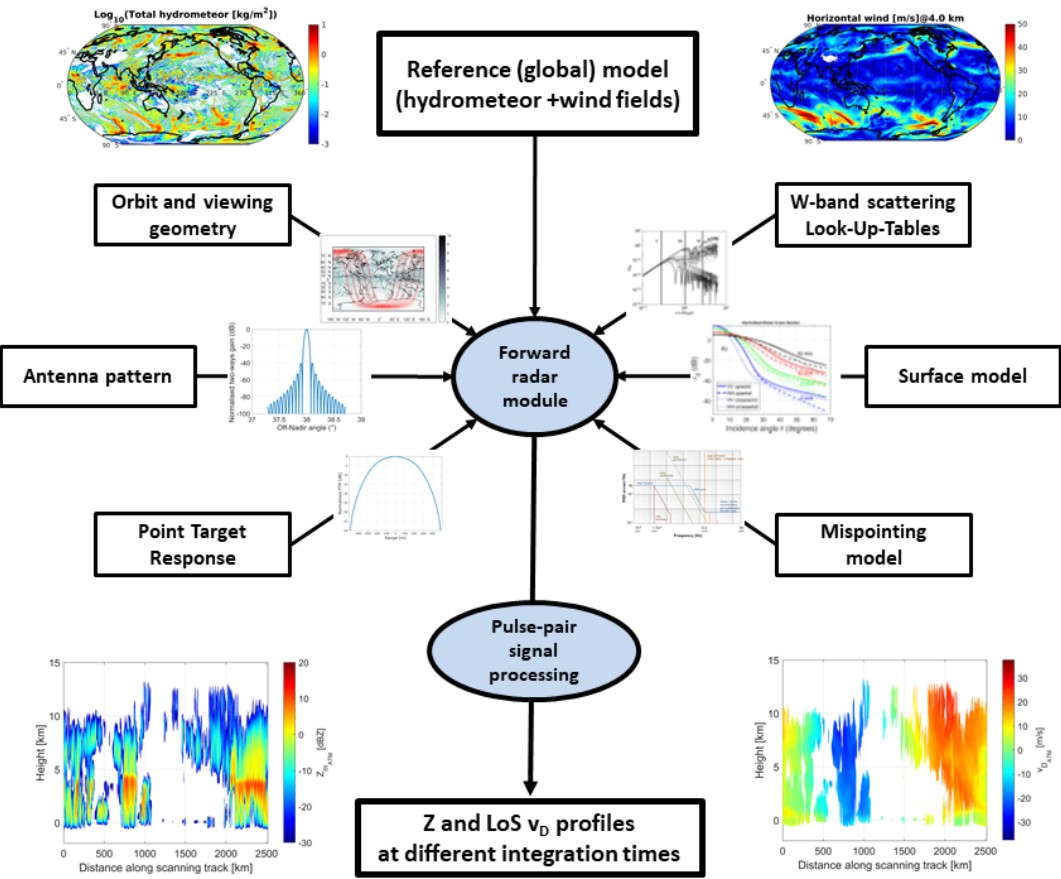

**Figure 2.** Flow chart illustrating the overall structure of the Wivern E2E simulator. The integrated hydrometeor content and the 4.0 km height winds are shown at the top as examples of input fields from the reference global model whereas outputs of the simulator (reflectivities and line-of-sight (LOS) winds) for a WIVERN "cross section" that will be examined later (Figs. 12-15) are presented in the bottom coloured panels.

configuration can be found in Stevens et al. (2019). The model output is available at the DYAMOND project web site via https://www.esiwace.eu/services/dyamond. The model outputs needed are temperature, pressure and relative humidity profiles plus the different hydrometeor water-equivalent contents. The different species are assumed to have different gamma size distributions (Testud et al., 2001). In principle any geo-located model that can produce such outputs can be ingested by the
5   simulator.

**Table 3.** Specifics of the radar for the simulation. The configuration here adopted is the one proposed for WIVERN in a recent ESA Earth Explorer 11 call. The E2E simulator can study various trade-offs to optimise mission, system and instrument parameters.

| | |
|---|---|
| Satellite altitude, $h_{sat}$ | 500 km |
| Satellite velocity, $v_{sat}$ | 7600 ms$^{-1}$ |
| off-nadir pointing angle | 38° |
| Incidence angle, $\theta_i$ | 41.6° |
| RF output frequency | 94.05 GHz |
| Pulse width | 3.3 $\mu$s |
| Antenna beamwidth, $\theta_{3dB}$ | 0.071° |
| Circular antenna diameter | 3 m |
| Rotation speed | 12 rpm |
| Footprint speed | 500 kms$^{-1}$ |
| Transmit polarization | H or V |
| Cross-polar isolation | <-25 dB |
| Single pulse sensitivity | -18 dBZ$^{\dagger}$ |
| H-V Pair Repetition Frequency | 4 kHz |
| Range sampling distance (rate) | 100 m (1.5 MHz) |
| Number of H-V Pairs per 1 km integration length | 8 |

$^{\dagger}$ A value of -15 dBZ maybe assumed to allow for a 3 dB margin.

## 2.2 Forward radar module

### 2.2.1 Orbital model and scanning geometry

The orbit selected for WIVERN is sun-synchronous with a mean inclination of 97.4°, a mean eccentricity of 0.001257, a mean local time of the ascending node equal 6:00 and 15 + 1/5 orbits per day which provides global coverage up to $\pm$ 82° latitudes. An example of the simulation of five orbits is shown in Fig. 3. By running several orbits it is possible to compute for each location the mean and maximum (i.e. the worst case scenario) revisit time of the WIVERN radar footprint; the latter is plotted as a function of latitude and longitude in the left panel of Fig. 4. The maximum revisit time has a strong latitudinal behaviour with a minimum in the equatorial band (peaking at more than 5 days) and a secondary peak at $\sim \pm 46°$ (exceeding 3 days at some longitudes). The maximum (blue line) and mean (red line) revisit time averaged over all longitudes as a function of latitude are shown in the right panel of Fig. 4. While the maximum revisit time presents different local maxima, the mean revisit time is monotonically decreasing from the Equator to the Poles with a mean value of 1.5 days in the Tropical band and of less than 1 day above 50° latitude, which leads to an average global revisit time of once a day between $\pm$82° latitude.

The radar is sounding the atmosphere down to the ground with a range resolution of 500 m. Fig. 5 and Fig. 6 illustrate the observing slant geometry. The actual vertical resolution will be the result of the slant range resolution, the antenna beamwidth

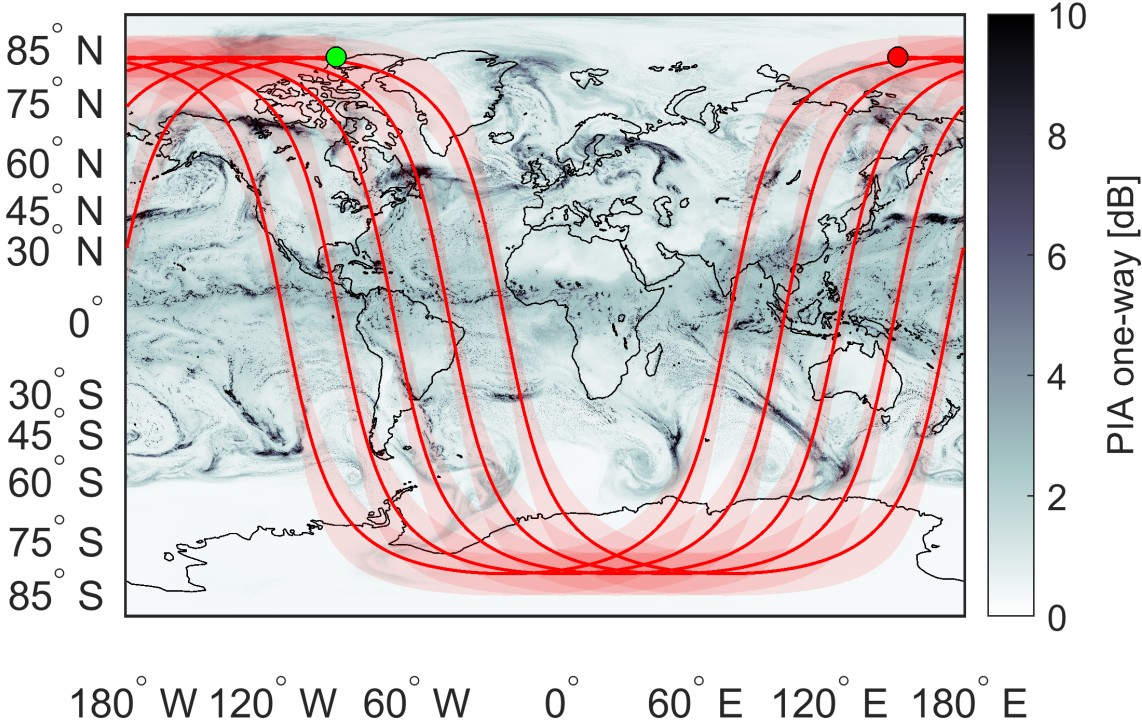

**Figure 3.** Example of simulation of five WIVERN orbits with the ground tracks (red lines), the 800 km WIVERN scanning swath (red-shaded region) plotted over the hydrometeor one-way path integrated W-band attenuation (the colorbar scale is in dB). A single model snapshot is used for the simulation.

and the satellite altitude (Meneghini and Kozu, 1990). Note that, for a uniform cloud, 90% (99%) of the backscattering power is coming from a region whose vertical extent is 640 m (980 m). The horizontal sampling pattern is a function of the rotation speed. The values used here (Tab. 3) are the result of a preliminary optimization for wind product performance (sensitivity and spatial resolution).

### 2.2.2 W-band scattering Look Up Tables

Scattering properties (extinction and backscattering coefficients, single scattering albedo, asymmetry parameters and Doppler velocities) at each model grid point are computed by adding up the contributions from the different hydrometeors (cloud water, cloud ice, rain, snow). Gas attenuation is computed according to the Rosenkranz (1998) model. The total scattering, backscat-

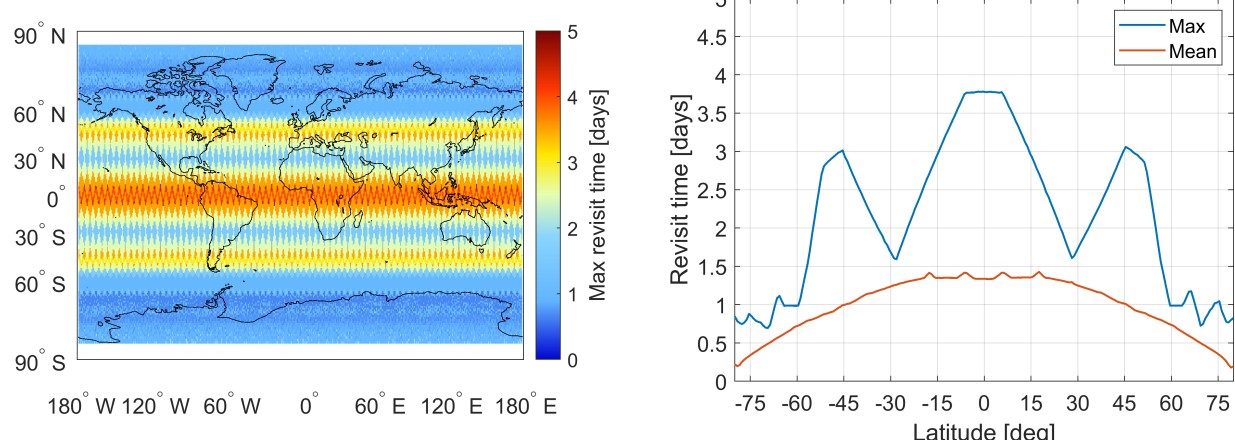

**Figure 4.** Left: WIVERN maximum (i.e. the worst case scenario) revisit time for a 15 + 1/5 day orbit. Right: the red dotted and blue continuous lines correspond to the latitude-averaged mean and maximum revisit time, respectively, as a function of latitude. Note a mean revisit time of 1.5 days in the Tropical band and of less than 1 day above $50°$ latitude. Globally, the mean revisit time is roughly daily.

tering and extinction coefficients are derived by adding up the single-particle scattering, backscattering and extinction cross sections for the different hydrometeor species according to their particle size distributions. Since all particle size distributions in the model are gamma size distributions, scattering properties are tabulated per unit mass concentration as a function of the mean mass weighted diameter and of the $\mu$ parameter (and of temperature, in case of liquid hydrometeors) like in the Appendix

of Battaglia et al. (2020b). Mie theory (Lhermitte, 1990) is used to compute the single-particle scattering properties. The class "Snow" (which represents all large ice particles) is assumed to have a constant density of 0.1 g/cm$^3$ with refractive index computed according to Maxwell-Garnett mixing formula (Kneifel et al., 2020). An exponential drop size distribution ($\mu = 0$) is assumed both for rain and snow with $N_0 = 8 \times 10^6\ m^{-4}$ (Marshall and Palmer, 1948) and $N_0 = 10^8\ m^{-4}$, respectively. The single scattering albedo is just the ratio between the scattering and the extinction coefficients whereas the asymmetry parameter

is derived as a weighted average of the different species asymmetry parameters with the scattering coefficients as weights. Currently the simulator only accommodate ice particles with fixed ice densities and hydrometeors with spherical shapes. The first issue can be resolved by changing the assumptions and switching the reference scattering Look-Up-Table. On the other hand the inclusion of preferentially oriented hydrometeors and dichroic media, which requires a polarization-dependent treatment of scattering and extinction (Battaglia et al., 2010a), is more complex. Such depolarization effects are not deemed to be as large

at W-band as at lower frequencies but they may be important by producing measurable differential phase shifts in ice clouds (Myagkov et al., 2020) and in deteriorating the Doppler velocity estimates by introducing decorrelation between the closely separated H and V-polarized pulses adopted with polarization diversity (Wolde et al., 2019). The inclusion of polarimetric variables is planned as future development (see Table 2) and will allow to compute parameters (Bringi and Chandrasekar, 2001)

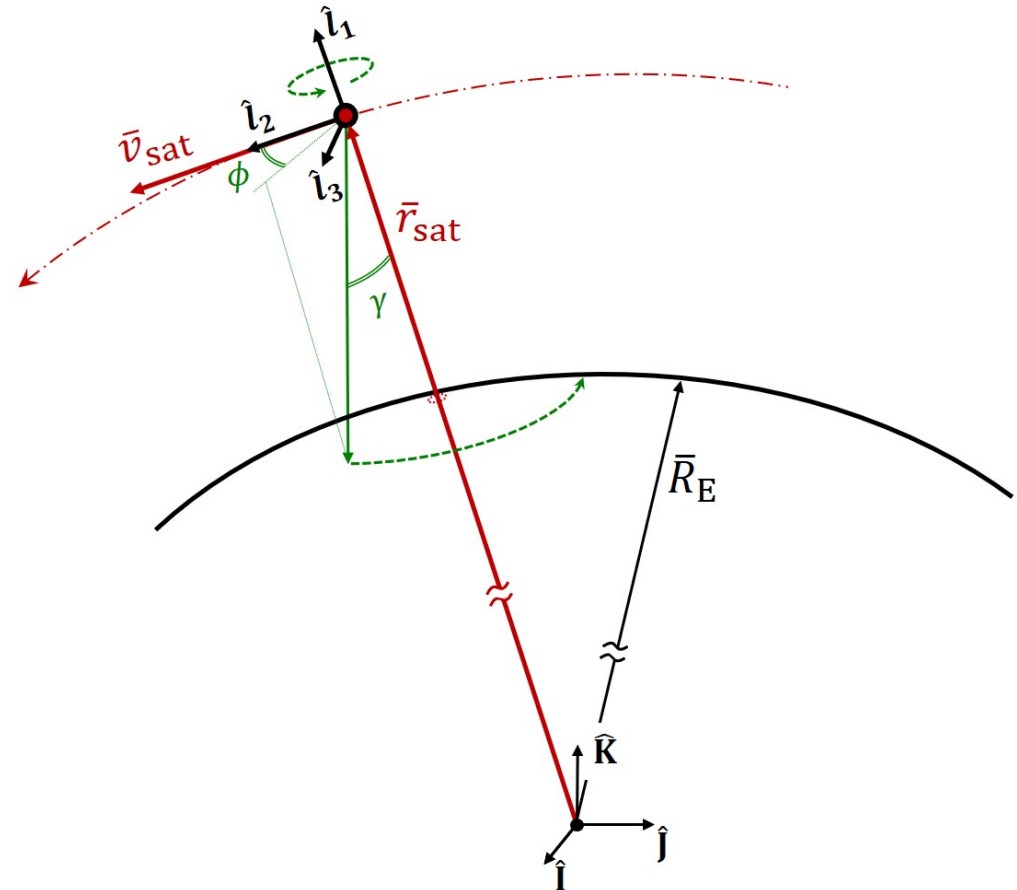

**Figure 5.** Illustration of the satellite scanning geometry. The boresight direction (solid green arrow) is identified by the elevation angle $\gamma = 38°$ with respect to the Nadir direction and by the azimuth angle $\phi$ measured from the horizontal direction $\hat{l}_2$ of the Local-Vertical/Local-Horizontal (LVLH) reference frame. The solid red arrows $r_{\text{sat}}$ and $v_{\text{sat}}$ represent the satellite's position and velocity vectors in the geocentric-equatorial (IJK) reference frame.

like differential attenuation, $A_{DP}$, reflectivity differential ratio, $Z_{DR}$, cross correlation coefficient, $\rho_{hv}$, and phase differential shift, $K_{DP}$.

In order to simulate the cross-polar reflectivities linear depolarization ratios ($LDR$) values are assigned to the different hydrometeor species based on $LDR$ climatology collected at the Chilbolton observatory (see Battaglia et al. (2018)). The different hydrometeors of the model ouput are assigned $LDR$ values drawn from a normal distribution with 1.5 dB standard deviation and mean values of -21, -19, -19 and -30 dB for rain, ice crystals, snow and cloud, respectively. $LDR$s in the assumed melting layer (at temperatures between the -1°C and +4°C isotherm) are assumed to have a mean value of -14 dB and a standard

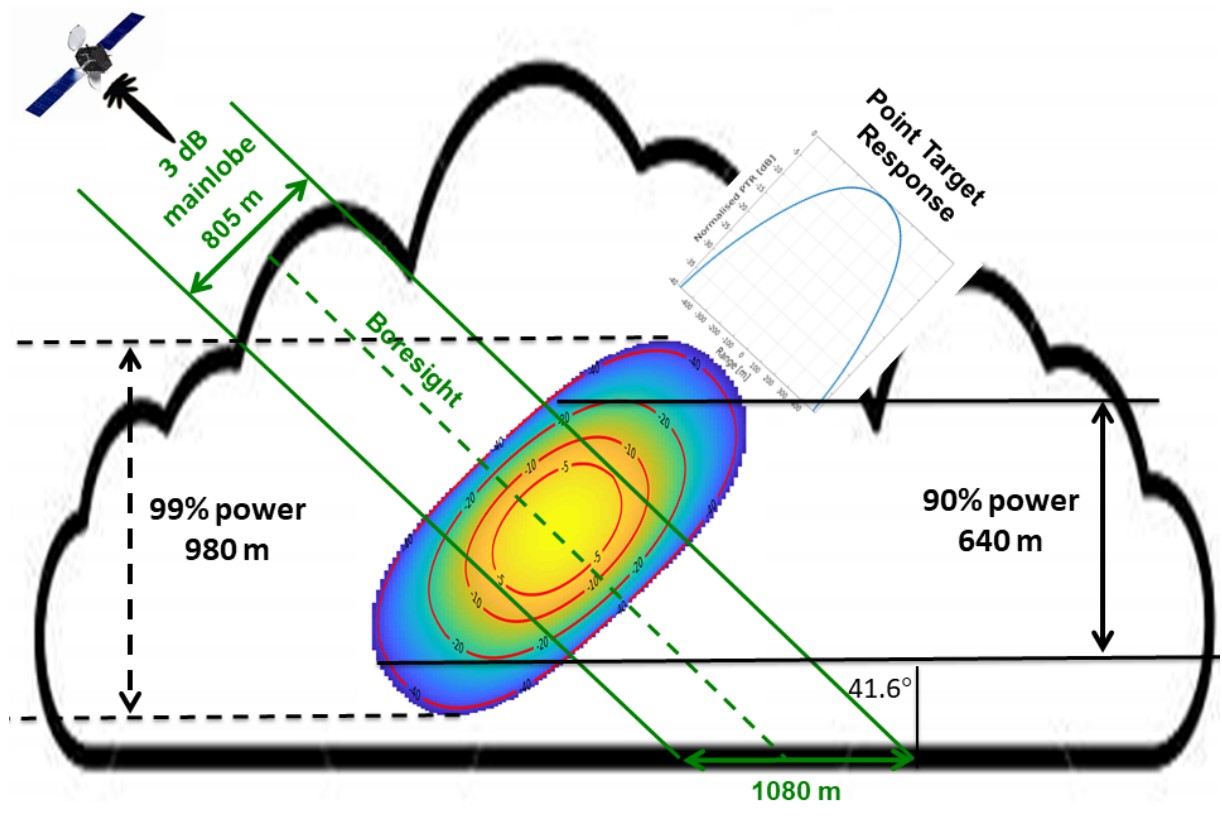

**Figure 6.** Schematic illustrating the 2D projection onto the antenna elevation cut of the WIVERN observing geometry. The specifics of the radar are detailed in Tab. 3.

deviation of 1.5 dB. Note that the 5°C range of temperature allowed for the melting layer generally tends to overestimate the thickness of the melting layer, thus the impact of the melting layer induced cross-talk can be overestimated. The $LDR$ values are only relevant when considering the cross-talk effects; at this stage we believe this approach is sufficient to demonstrate what is the climatological impact of the ghosts in worsening Doppler velocity precisions (Sect. 2.3.2).

5 **2.2.3  Surface model**

The normalised surface backscattering cross sections ($\sigma_0$, Meneghini and Kozu (1990)) are assumed to be normally distributed around -25 dB and -8 dB for sea and land respectively with 3 dB standard deviation whereas the surface $LDR$ is assumed to be -14 dB and -6 dB for sea and land with 1 dB standard deviation (Battaglia et al., 2017). In case of coastal regions a weighted mean accounting for the surface type fraction is taken.

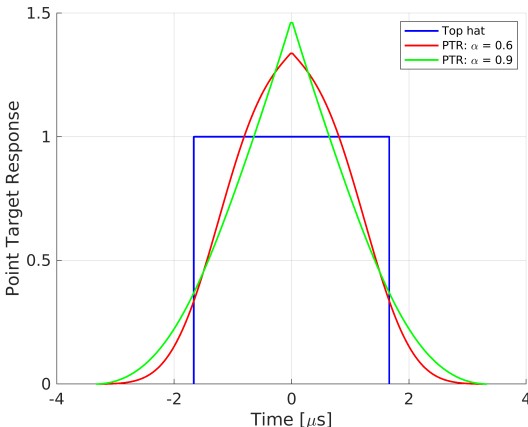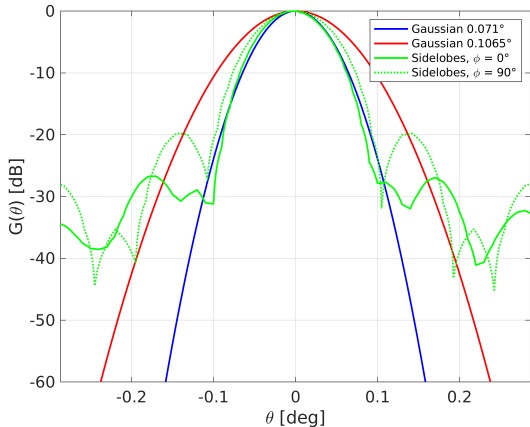

**Figure 7.** Left panel: examples of point target responses that can be used as inputs in the simulator. The narrow top hat (blue curve) is the one adopted for the simulations later on. Right panel: examples of antenna patterns that can be used as inputs in the simulator. The narrow Gaussian one (blue curve) is the one adopted for the simulations later on. The green ones correspond to the elevation and azimuthal cut of an antenna pattern for an elliptical antenna with sidelobes.

#### 2.2.4 Point target response

The point target response (PTR) is assumed to be a simple top hat with a pulse length, $\tau_p$, of $3.3\mu$s. Correspondingly, the range resolution becomes $\Delta r = c\tau_p/2$. More sophisticated PTR function could be used in order to optimise the equivalent noise bandwidth and PTR width (left panel in Fig.7). The PTR is used as convolution function along range for all the radar observables.

#### 2.2.5 Antenna pattern

Since the WIVERN antenna is circular a simple Gaussian antenna pattern is assumed with a one-way gain equal to:

$$G(\theta_a) \quad = \quad G_0 \exp\left[-4 \log(2)\left(\frac{\theta_a}{\theta_{3\text{dB}}}\right)^2\right] \equiv G_0\, f_a(\theta_a), \tag{1}$$

where $G_0$ is the antenna gain in the boresight direction, $\theta_a$ is the antenna polar angle with respect to the boresight and $\theta_{3\text{dB}}$ is the antenna 3-dB beamwidth. Any antenna pattern inclusive of side lobes can be added by simply sampling it on the angles used later on for the solid angle integration (right panel in Fig.7).

### 2.3 Simulation of polarization diversity radar observables

The Doppler velocity in radar systems is derived by measuring phase shifts between successive pulses (pairs). Since phases are measured with a $2\pi$ periodicity, this methodology introduces an ambiguity with a folding Nyquist velocity equal to $v_{Ny} =$

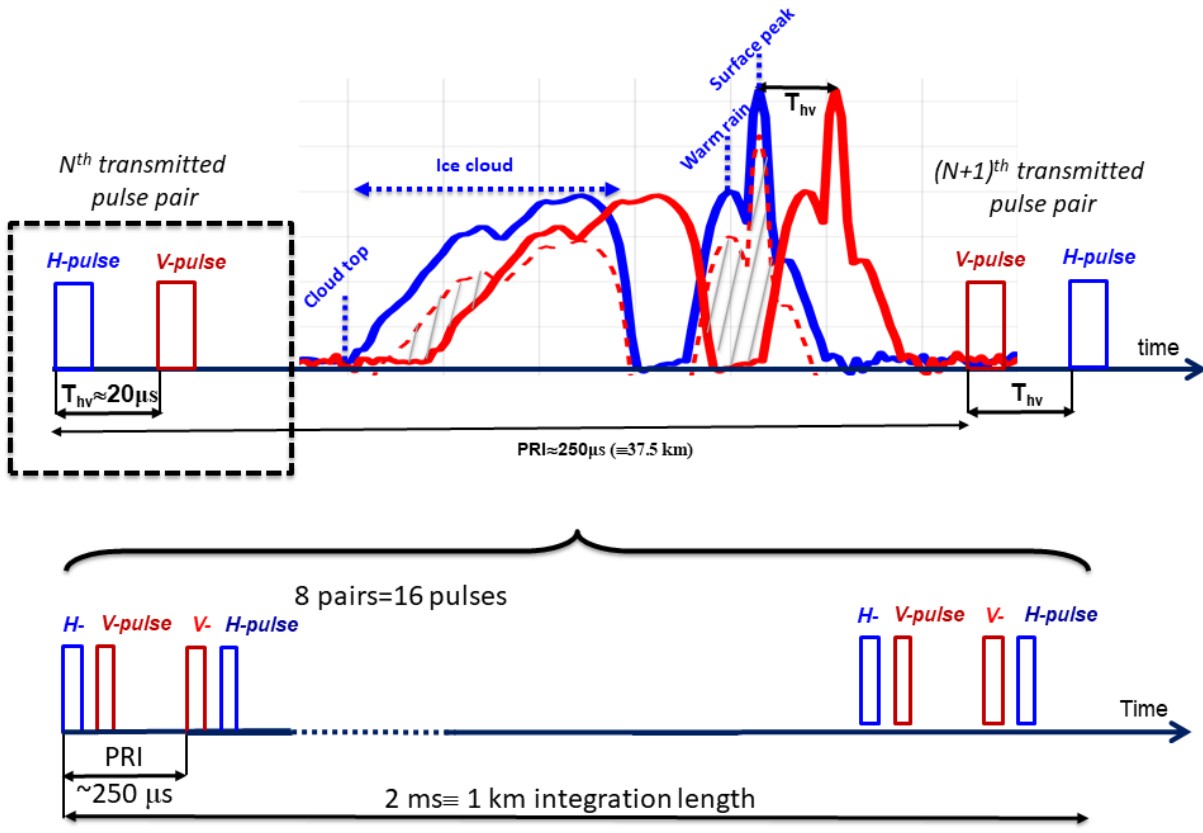

**Figure 8.** Bottom panel: the timing of the transmitted pulse sequence proposed for WIVERN with interlaced H-V pairs. A sequence of $M = 8$ pairs correspond to 2 ms, equivalent to a 1 km distance along the scanning track. Note that the order of the polarization state of each pulse pairs is switched from pulse to pulse in order to cancel out differential phase shift effects between the two channels. Top panel: example of the return echoes from a scene including an ice cloud, a cloud free region and warm rain above a strongly reflecting surface. The returns in the H channel are plotted in blue, those in the V channel (lagging by 20 $\mu$s) in red. The dashed red line corresponds to the interference caused by the blue H pulse encountering a depolarising target. A very high depolarization ratio of -10 dB has been used to exacerbate this effect that leads to returns in regions void of hydrometeors (later referred to as "ghosts") in the red H channel. The hatched areas represent ranges where the "ghosts" exceed the co-polar signal. In this case the one from the ground and the warm rain is much more serious (and appears shifted upward by circa 3 km in correspondence to the cloud free region) than the one caused by the large Z gradients at the top of the cloud. A similar reasoning applies to the H-channel (not shown for clarity of purpose).

$\pm \frac{\lambda}{4PRI}$. This issue could be mitigated by reducing the pair repetition interval (PRI). However, this has the drawback of decreasing the maximum unambiguous range ($r_{max} = c\,PRI/2$) and, in space-borne applications, can actually significantly reduce the correlation between pulses (thus undermining the Doppler methodology). In order to solve this "range/correlation–Doppler dilemma" the pulse scheme illustrated in Fig. 8 has been proposed (Pazmany et al. (1999); Kobayashi et al. (2002); Battaglia

et al. (2013)). Horizontally and vertically polarised pulses are sent out with a short time separation (indicated as $T_{hv}$ in the diagram) with relatively low repetition frequency; this effectively decouples the maximum unambiguous range from the Nyquist velocity because the H and V pulses propagate, backscatter and can be received almost independently.

WIVERN will transmit pairs of 3.3 $\mu$s long H- and V-polarised pulses with a separation of $T_{hv} = 20$ $\mu$s at a pulse repetition frequency of 4 kHz (Fig. 8). This parameter selection corresponds to $v_{Ny} = \pm 37.5$ ms$^{-1}$, sufficiently high for unfolding the highest winds, and to $r_{max} = 37.5$km.

The fundamental radar quantities are the range dependent $I$ and $Q$ time series. The simulation of $I$ and $Q$s for a system adopting polarization diversity is described in Battaglia and Kollias (2015). Here we are interested in the Level 1 radar observables: reflectivities and Doppler velocities. Thus we adopt a simpler approach and use theoretical results to derive the noisiness of the reflectivity and velocity fields. Both the volume scattering from the atmosphere and the surface scattering from the ground-return must be accounted for when computing such observables.

### 2.3.1 Simulation of reflectivities

The power received by the radar from the atmosphere, $P_r^{atm}(t)$ is given by an integral over the backscattering volume (Bringi and Chandrasekar, 2001):

$$P_{rec}^{atm}(t) = P_{tr} \frac{G_0^2 \lambda^2}{(4\pi)^3} \int_{\phi_a=0}^{2\pi} \int_{\theta_a=0}^{\pi} f_a^2(\theta_a) \int_0^\infty \frac{\eta(r,\theta_a,\phi_a)}{r^2} |PTR(t-2r/c)|^2 \ e^{-2\int_0^r k_{ext}(s)ds} \ dr \ d\Omega_a \tag{2}$$

where $\eta$ is the radar reflectivity or backscattering cross section per unit volume, $P_{tr}$ is the transmitted power, $\lambda$ is the wavelength of radar, $k_{ext}$ is the extinction coefficient, $\phi_a$ is the azimuthal angle in the antenna reference system and $d\Omega_a = \sin\theta_a \, d\theta_a \, d\phi_a$ is the infinitesimal antenna solid angle. The equivalent radar reflectivity factor, $Z$, is the quantity that is generally used in meteorology instead of $\eta$. It is defined as:

$$Z = \frac{\lambda^4}{\pi^5 |K_w|^2} \eta \tag{3}$$

where $|K_w|^2$ is the dielectric factor for water (Bringi and Chandrasekar, 2001). In the following we assume the convention to set $|K_w|^2 = 0.93$. Practically in order to compute the reflectivity factor corresponding to the atmosphere the three dimensional integral in Eq. (2) is first broken into an integral over the solid angle (defined with respect to the boresight direction); this allows computing $Z$ for ranges $r_i$ sampled at distance $\delta r$ (=100 m in our case but adjustable to the specific need):

$$Z_{\delta r}^{atm}(r) = \frac{\int_{\phi_a=0}^{2\pi} \int_{\theta_a=0}^{\pi} f_a^2(\theta_a) Z_e(r,\theta_a,\phi_a) \ e^{-2\int_0^r k_{ext}(s)ds} \ d\Omega_a}{\int_{\phi_a=0}^{2\pi} \int_{\theta_a=0}^{\pi} f_a^2(\theta_a) \ d\Omega_a} \equiv \frac{\int_{\phi_a=0}^{2\pi} \int_{\theta_a=0}^{\pi} f_a^2(\theta_a) Z_e(r,\theta_a,\phi_a) \ e^{-2\int_0^r k_{ext}(s)ds} \ d\Omega_a}{\Omega_{2A}}$$

$$\tag{4}$$

where $\Omega_{2A}$ is the two-way antenna main-lobe solid angle (equal to $\pi\theta_{3dB}^2/(8\ln 2)$ for a Gaussian antenna). The solid angle integral is performed by sampling 7 polar and 21 azimuthal angles with respect to the antenna boresight by trapezoidal integration.

Then $Z_{\delta r}^{atm}(r)$ is convolved with the point target response:

$$Z^{atm}(r) = w_{PTR} * Z_{\delta r}^{atm}(r) \tag{5}$$

where $w_{PTR}$ is the normalised point target response.

The power received by the radar from the surface at a range $r$, $P_r^{surf}(r)$ is computed by an integration performed over the surface, $\Sigma$, which is obtained from the intersection between the surface and the spherical shell with radius between $r - \Delta r/2$ and $r + \Delta r/2$ with $\Delta r = c\tau_p/2$ (Meneghini and Kozu, 1990):

$$P_{rec}^{surf}(r) = P_{tr}\frac{G_0^2\lambda^2}{(4\pi)^3}\int\limits_{\Sigma}\frac{\sigma_0(\theta_a,\phi_a)\,f_a^2(\theta_a)\,e^{-2\int_0^r k_{ext}(s)ds}}{r^4}\,d\Sigma \equiv P_{tr}\frac{G_0^2\lambda^2}{(4\pi)^3}\mathcal{I}_{surf}(r) \tag{6}$$

where $\sigma_0$ is the normalised radar cross section (NRCS). The surface contribution can be written as an equivalent reflectivity term as:

$$Z^{surf}(r) = \frac{\lambda^4}{\pi^5|K|^2}\frac{1}{\Omega_{2A}}\frac{r^2}{\Delta r}\mathcal{I}_{surf}(r) \tag{7}$$

The integral $\mathcal{I}_{surf}$ defined in Eq. (6) is evaluated by numerical integration on a 3 km$^2$ grid defined on the plane tangent to the Earth at the intersection between the Earth and the antenna boresight. Eqs. (6-7) have been applied to a $\delta r$ (=100 m) smaller than $\Delta r$ (500 m) to compute $Z_{\delta r}^{surf}(r)$, similarly to what has been done in Eq. (2). Then $Z^{surf}(r)$ can be computed with a formula analogous to Eq. (5).

The total reflectivity signal is obtained by adding up the atmospheric and the surface contributions, e.g. for the V-channel:

$$Z_{VV}(r) = Z_{VV}^{surf}(r) + Z_{VV}^{atm}(r). \tag{8}$$

Both are saved in order to compute the impact of the clutter on the radar observables at low altitudes.

To simulate a Doppler radar with polarization diversity profiles cross-polar returns are also needed. These are obtained performing the same integrals but using the cross-polar reflectivities via $LDR$ and the cross polar surface NRCS, $\sigma_0^{HV}$. The cross-polar reflectivities will be important to compute the appearance of the "ghosts" (Battaglia et al., 2013; Illingworth et al., 2018b; Wolde et al., 2019). The reflectivity signal received in the V-channel, $Z_V$, is the combination of the co-polar V-signal, $Z_{VV}$ (continuous red line) combined with the anticipated cross talk of the H-signal, $Z_{HV}$ (dashed red line):

$$Z_V(r) = Z_{VV}(r) + Z_{HV}(r + cT_{hv}/2) \tag{9}$$

The hatched regions in the top panel of Fig. 8 highlight the ranges where the cross-signal exceed the copolar signal and therefore will significantly modify the reflectivity signal.

Similarly the signal received in the H-channel, $Z_H$, is the combination of the co-polar H-signal, $Z_{HH}$, (continuous blue line) combined with the delayed cross talk of the V-signal, $Z_{VH}$, (not shown):

$$Z_H(r) = Z_{HH}(r) + Z_{VH}(r - cT_{hv}/2) \tag{10}$$

The order of the polarization state of each pulse pairs is switched from pulse to pulse (see bottom panel in Fig. 8) in order to cancel out differential phase shift during propagation between the radar and the targets and for any difference in the lengths of the two polarization transmission lines (Pazmany et al., 1999). Therefore, if we assume no differential reflectivity ($Z_{HH} = Z_{VV} = Z_{co}$), reciprocity ($Z_{HV} = Z_{VH} = Z_{cx}$) and the same gain in the two linearly polarized channels, after integration of $M$-pairs of pulses ($M = 8$ in the bottom panel of Fig. 8) what is practically measured is:

$$Z_1(r) = Z_{co}(r) + Z_{cx}(r + cT_{hv}/2) \tag{11}$$

in the co-polar channel after integrating the first pulses of each of the $M$ pairs and

$$Z_2(r) = Z_{co}(r) + Z_{cx}(r - cT_{hv}/2) \tag{12}$$

after integrating the second pulses of the $M$ pairs.

Since the Doppler spectral widths, $\sigma_v$, are expected to exceed 3 ms$^{-1}$ for all scanning directions since the Doppler fading due to the satellite fading is equal to $\sigma_v = \frac{v_{sat_\perp} \theta_{3dB}}{4\sqrt{\ln 2}}$ where $v_{sat_\perp}$ is the component of the satellite velocity perpendicular to the boresight direction, we can consider reflectivity measurements, separated by a pulse repetition interval ($PRI$), as independent (for instance the correlation function for 3 ms$^{-1}$ and a time lag equal to 250 $\mu s$ is 0.0072). Therefore the number of independent samples practically is identical to the number of samples. For each single pulse we simulate the total power $P$ as a combination of noise, $N$ (equal to -18 dBZ) and signal, $S$ (equal to the expressions given in (11-12)) by using the fact that the probability distribution of power is a simple exponential with a standard deviation equal to the mean (Doviak and Zrnić, 2006), i.e.:

$$P_{single\, pulse} = -\log(\xi)(N + S) \tag{13}$$

where $\xi$ is a random number uniformly distributed between 0 and 1. Note that, since we oversample in range every 100 m, the application of Eq. (13) must be performed before the convolution in range (Eq. 5) because oversampled reflectivities and Doppler velocities are not independent. Power is averaged along track by simply averaging single pulses powers. Since the WIVERN footprint moves at about 500 kms$^{-1}$, 8 pulses must be averaged per km for each of the two channels (Fig. 8).

### 2.3.2 Doppler variables

The radar Doppler velocities also have a component associated with the hydrometeor and one with the surface. The former is given by:

$$v_D^{atm}(r) = \frac{\iiint_V v_{LOS}^{atm} Z_{co} G^2 dV}{\iiint_V Z_{co} G^2 dV} \tag{14}$$

where $V$ is the backscattering volume (coloured region in Fig. 6), $v_{LOS}^{atm}$ is the projection of the satellite velocity minus the hydrometeor velocity (the result of the wind speed and the hydrometeor fall-speed) along the line of sight (LOS) and $Z_{co}$ is the co-polar reflectivity factor. Note that the ghosts echoes will have random phase, so will not produce any bias in the wind but only a loss of precision (Pazmany et al., 1999; Battaglia et al., 2018; Wolde et al., 2019). NUBF effects (Battaglia and Kollias,

2015; Battaglia et al., 2018) can be assessed by setting in Eq. (14) the satellite velocity equal to zero and looking at the change from the Doppler velocities computed with the actual satellite velocity.

Similarly to Eq. (6) the Doppler velocity associated with the surface will be equal to:

$$v_D^{surf}(r) = \frac{\int_\Sigma \frac{v_{LOS}^{surf}\, \sigma_0(\theta_a,\phi_a)\, f_a^2(\theta_a)\, e^{-2\int_0^r k_{ext}(s)\,ds}}{r^4}\, d\Sigma}{\int_\Sigma \frac{\sigma_0(\theta_a,\phi_a)\, f_a^2(\theta_a)\, e^{-2\int_0^r k_{ext}(s)\,ds}}{r^4}\, d\Sigma} \tag{15}$$

where $v_{LOS}^{surf}$ is projection of the the satellite velocity onto the line of sight. Here we assume that the surface is still but any movement could be added if, for instance, ocean currents were available.

Doppler velocities estimated via pulse-pair processing also have intrinsic noise associated with the phase and thermal noise and to the cross-polarization interference. Uncertainties depend on the signal to noise ratio ($SNR$), the radar Doppler spectral width and the number of averaged samples (Battaglia et al., 2013; Illingworth et al., 2018a). Following Pazmany et al. (1999), the estimate of the variance of the mean Doppler velocity for $M$ independent pulse pair samples can be written as:

$$var_{\hat{v}_D} = \frac{1}{M}\frac{v_{Ny}^2}{2\pi^2\beta^2}\left[\left(1+\frac{1}{SNR}\right)^2 + \frac{1}{SGR_1} + \frac{1}{SGR_2} + \frac{1}{SGR_1\times SGR_2} + \frac{1}{SNR\times SGR_1} + \frac{1}{SNR\times SGR_2} - \beta^2\right] \tag{16}$$

$$\beta^2 \equiv e^{-\frac{16\pi^2\sigma_v^2 T_{hv}^2}{\lambda^2}}; \quad SNR = \frac{S}{N}; \quad SGR_1 = \frac{Z_{co}(r)}{Z_{cx}(r-cT_{hv}/2)}; \quad SGR_2 = \frac{Z_{co}(r)}{Z_{cx}(r+cT_{hv}/2)}. \tag{17}$$

where we have introduced definitions for the signal to noise ($SNR$) and signal to ghost ($SGR$) ratios. A Gaussian random noise with standard deviation corresponding to Eq. (16) is added to the velocities, which are then folded back into the Nyquist interval, $v_{Ny} = \frac{\lambda}{4T_{hv}}$.

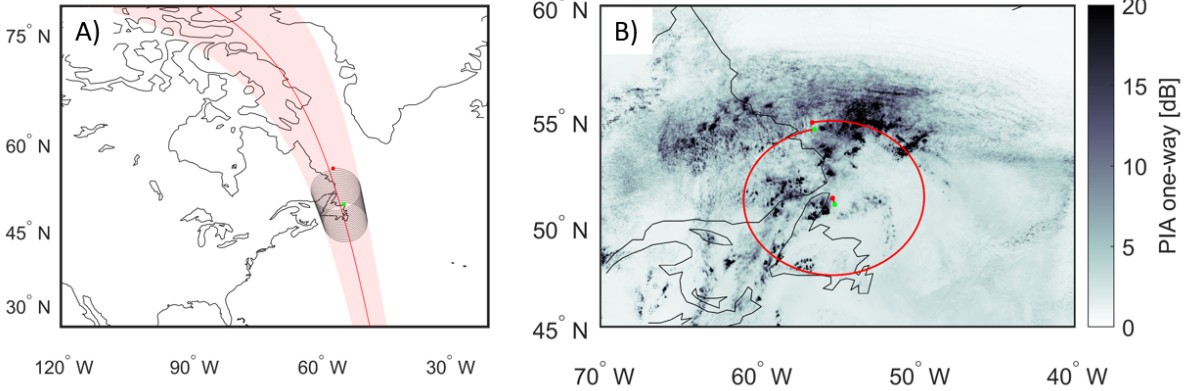

**Figure 9.** Panel A: WIVERN satellite track off the Labrador coast with the satellite ground track (red line), the scanning swath (shaded red region) and the radar footprints (black line) for 20 full revolutions of the conically scanning antenna corresponding to a flight time of 100 s. Panel B: details of a single revolution of the WIVERN antenna with the one way path integrated attenuation due to the hydrometeors shown in the background. This rotation sample will be examined in detail later (see Figs. 12-15).

## 2.4 Mispointing modelling

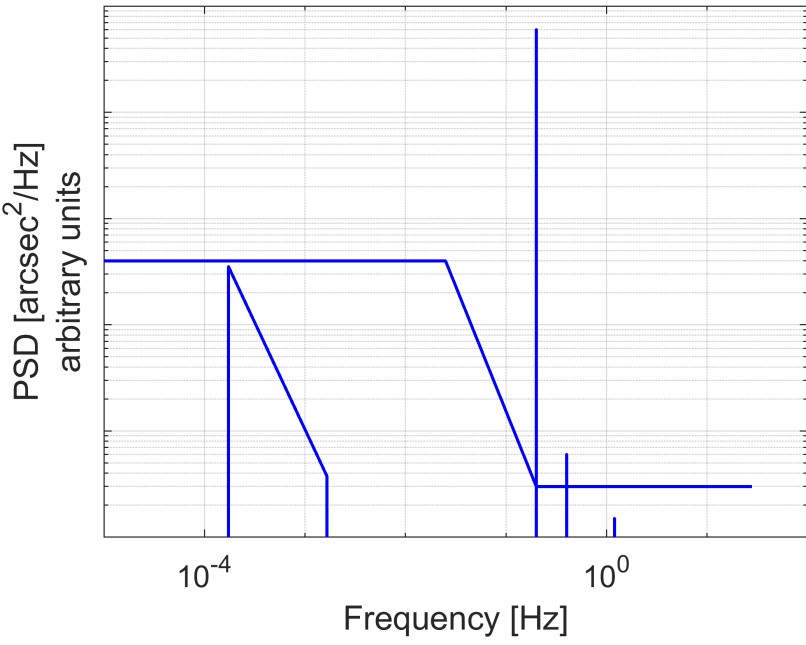

**Figure 10.** Conceptual model of the azimuth absolute knowledge error PSD. Contributions from different mechanisms are expected in different regions of the spectrum. For instance, sharp peaks are expected in correspondence to the scan harmonics.

For accurate winds the pointing of the radar beam formed by the antenna must be known very accurately. For instance, a 140 $\mu$rad uncertainty in elevation or azimuth angles or either can potentially lead to a 1.0 ms$^{-1}$ LOS wind uncertainty. The antenna boresight direction can be identified by two angles: the elevation and the azimuthal angle (see Fig. 5). The former can be monitored by controlling the sea surface return range whereas the knowledge of the azimuthal angle is more challenging. The azimuth mispointing is usually described in terms of its frequency distribution by a Power Spectral Density (PSD). A previous industrial study conducted for the SKIM mission (Ardhuin et al., 2019) predicts a PSD with a low frequency (orbit to seasonal scale) component dominated by the satellite stability and the antenna Thermo Elastic Distorsion (TED) and a high frequency component affected by antenna and satellite micro-vibrations. A schematic PSD for the azimuthal angle mispointing is sketched in Fig. 10. PSDs provide the input for our simulator. Time series representations of mispointing angles, $\Delta\phi$, can be produced by firstly constructing a frequency domain signal and then applying the Inverse Fast Fourier Transform (IFFT). The one-sided PSD in Fig. 10 is sampled at discrete frequencies, going from zero to the Nyquist critical frequency $f_c$. The one-sided PSD is then mirrored into a two-sided power spectrum. Since the total power must be preserved, the values in the two-sided PSD are half the values of the one-sided PSD, except for the ones associated with the frequencies 0 and $\pm f_c$.

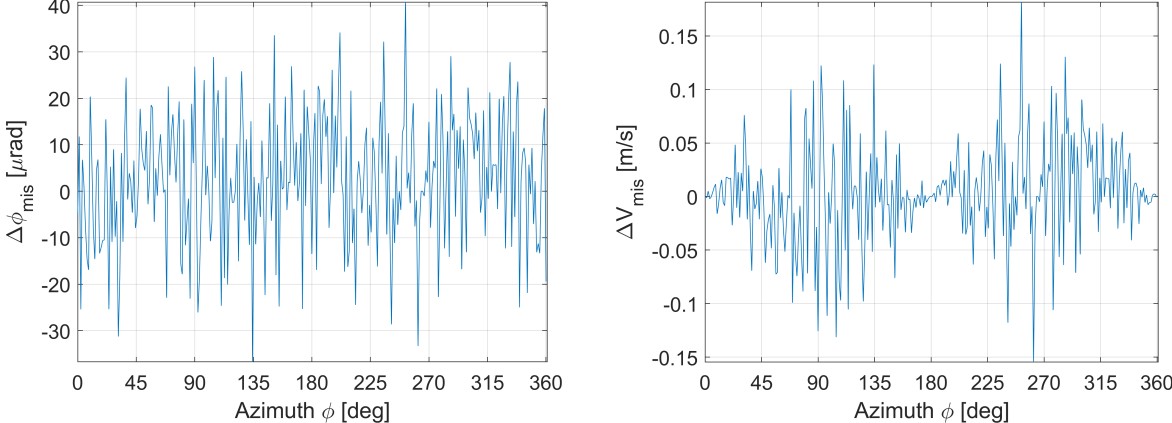

**Figure 11.** Left panel: One possible time series of the azimuthal scanning angle mispointing corresponding to a PSD like in Fig. 10 as provided by preliminary industrial studies. Right panel: LOS velocity error corresponding to the mispointing shown in the left panel. Mispointing errors corresponding to the PSD in Fig. 10 are generally smaller than $0.1$ ms$^{-1}$, far lower than the precision of the Doppler velocity observations.

The amplitude of the two-sided spectrum of the signal is calculated from the two-sided PSD by taking the square root and adding to each sample a random phase in the $[0, 2\pi]$ interval. The spectrum is forced to be conjugate symmetric, so that the IFFT returns a real-valued time series for the mispointing angle. An example of such a time series for a single antenna revolution is shown in Fig. 11 (left panel) with the corresponding LOS velocity error (right panel). The amplitude of the velocity error is a strong function of the azimuthal position. If $\phi$ is the azimuthal angle measured clockwise from the forward direction then the error can be approximated as:

$$\Delta v_{mis} = v_{sat} \sin(38°) \left[ \frac{1}{2} \cos(\phi) \, \Delta\phi_{mis}^2 + \sin(\phi)\Delta\phi_{mis} \right] \approx v_{sat} \sin(38°) \sin(\phi)\Delta\phi_{mis}, \tag{18}$$

which clearly shows that the error is minimised close to the forward and backward directions and amplified at side views. When inputting a realistic PSD as derived from initial industrial studies (internal communications, confidential) the error due to azimuthal mispointing remains always smaller than $0.17$ ms$^{-1}$, thus it will provide a very small contribution to the Doppler velocity error budget.

## 2.5 Radiometric mode

WIVERN is also envisaged to have a radiometric mode. During the $250$ $\mu$s time between transmitted pulse pairs, there will be a dedicated time (of the order of 10%) with a dedicated receiver with broad bandwidth ($> 20MHz$) for each receiver. The brightness temperatures in the two polarization modes are simulated by an Eddington radiative transfer model (Kummerow, 1993) by using the slant one-dimensional approximation (Battaglia et al., 2005) and the scattering, extinction, asymmetry parameter and temperature profiles derived form the model outputs. Land emissivities are polarization independent and assumed

to be equal 0.9 whereas ocean emissivities are computed via the TESSEM model (Prigent et al., 2017) with the 10 m wind and the sea surface temperature from the model product. Preliminary assessments (Thales Alenia Space, personal communications) suggest that brightness temperature uncertainties at 5 km scale integration should be below 3 K.

## 3 Applications of the E2E simulator

### 3.1 Case study: system over Labrador

The simulator rationale is demonstrated for a case study simulating an overpass over Labrador on the 5th September 2017, with a cold front moving eastward from inland. The satellite is moving northward and is scanning counterclockwise. The satellite ground track over North America is shown in Fig. 9 with a detail of the scanning pattern shown only for the region off the Labrador coast (Panel A). A full scan circle (5 s) is simulated in detail and shown in Panel B. For this full scan circle, Fig. 12 shows the antenna weighted hydrometeor water content, $WC$, (Panel A) and LOS winds (Panel B) computed using the following equations:

$$
\begin{aligned}
WC_{AW}(r) &= \frac{\iiint_V WC\, G^2 dV}{\iiint_V G^2 dV}, \\
v_{AW}(r) &= \frac{\iiint_V v_{LOS}\, G^2 dV}{\iiint_V G^2 dV}.
\end{aligned}
\tag{19}
$$

The x-label "distance along the scanning track" used here and in the following figures corresponds to the length along the ground projection of the rotating antenna boresight with 2500 km corresponding to a 360° rotation. A variety of cloud and precipitation types is present in the scene with multiple layers of ice and liquid clouds at different heights and with disparate thicknesses. The LOS winds show a characteristic alternating sign behaviour associated with the conically scanning geometry and present strong vertical variability in some areas (e.g. in the lower troposphere).

Reflectivities and mean Doppler velocities for the atmospheric and surface targets computed according to the methodology described in Sect. 2.3.1 are shown in Fig. 13. The atmospheric reflectivity mirrors the hydrometeor contents but with a region of strong attenuation corresponding to heavy rain from 2000 km onward. Only reflectivities above -30 dBZ are shown. The surface reflectivity and Doppler velocities are shown in the bottom panels. The reflectivity of the surface is clearly modulated by two effects: atmospheric attenuation (very strong e.g. between 2050 and 2100 km in the along-track coordinate) and $\sigma_0$ variability with large discontinuities at sea-land transitions (e.g. at about 670, 805 and 1390 km in the along-track coordinate). Note that the clutter signal tends to decrease to very low levels (<-30 dBZ) at a height of 1 km. This confirms previous findings (Illingworth et al., 2020). However, attention should be paid in future work to antenna sidelobes that can effectively enhance clutter contamination on Doppler velocity signal especially over land (see Illingworth et al., 2020, Fig. 8).

The surface Doppler velocities, sampled at very fine range resolution (bottom right panel of Fig. 13) shows its characteristic behaviour with zero velocity at the surface range (the satellite velocity along the antenna boresight is always subtracted out) and a pattern of positive and negative velocities at other ranges with a strong dependence on the scanning azimuthal angle, which is used as an alternative x-axis coordinate in the bottom right panel. Note that the surface Doppler velocities are always

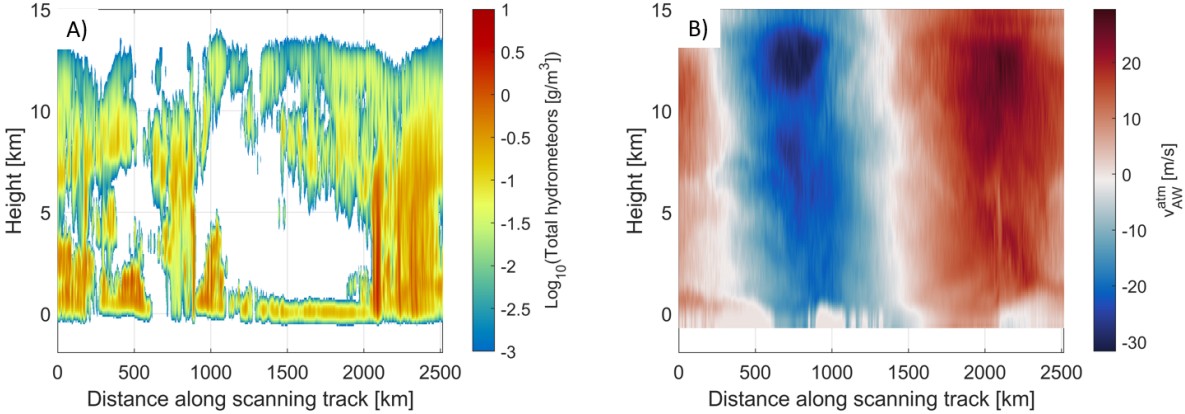

**Figure 12.** Antenna weighted hydrometeor content in g/m³ expressed in base-10 logarithmic scale (Panel A) and LOS winds (Panel B) in correspondence to the revolution shown in Fig. 9B. Only hydrometeor contents above 1 mg/m³ are shown. The change in velocity reflects the different antenna viewing directions of the weather system.

zero at the surface ranges because the surface is assumed to be still. The azimuthal angle is measured clockwise from the forward looking direction (where it is in the same direction as the satellite motion). When the radar is side-looking the surface appears perfectly still at all altitudes whereas when the radar is looking in the forward or backward directions there is a strong variability with altitudes. As a result, the bias in Doppler velocities induced by clutter contamination will depend on the signal to clutter ratio, the altitude and the azimuthal scanning direction. Overall, when averaging over heights and azimuthal angles, the clutter contamination will produce a bias towards zero Doppler velocities, i.e. the ground clutter will tend to mute the boundary layer winds.

The $LDR$ values shown in the left panel of Fig. 14 clearly have highest values in the melting layer and the land surfaces. These two regions are the major sources of ghosts as can be deduced by looking at the $SGR$ (right panel of Fig. 14), with strongly negative values associated with the ghosts generated by the surface at heights straddling $\pm 2.3$ km and with larger $SGR$s at about 6 km associated with the ghosts caused by the melting layer. Ghosts tend also to appear at cloud top (see strongly negative $SGR$s in such regions in the right panel of Fig. 14), a phenomenon which, if not accounted for, will tend to artificially thicken high clouds.

The two panels of Fig. 15 show simulations of WIVERN products: reflectivities (left) and LOS Doppler velocities (right panel) after 1 km along-track integration. The reflectivities ($Z(r) = 0.5 [Z_1(r) + Z_2(r)]$) are the averages of $M = 8$ pairs and include signal and noise. At such an integration length the sensitivity (after noise subtraction) is expected to be -22.5 dBZ, i.e. $5 \log_{10}(8) = 4.5$ dB better than the single-pulse sensitivity. Only regions with signals exceeding this level are plotted in Fig. 15.

The presence of ghosts arising from surface cross-talk is obvious around an altitude of $\pm 2.3$ km. Because of the considerably higher $\sigma_0^{HV}$ over land, the signal to ghost ratios $SGR$s defined as the minimum values between $SGR_1$ and $SGR_2$ defined in Eq. (17), are significantly smaller over ocean than over land, where they almost disappear below the noise level (right panel of

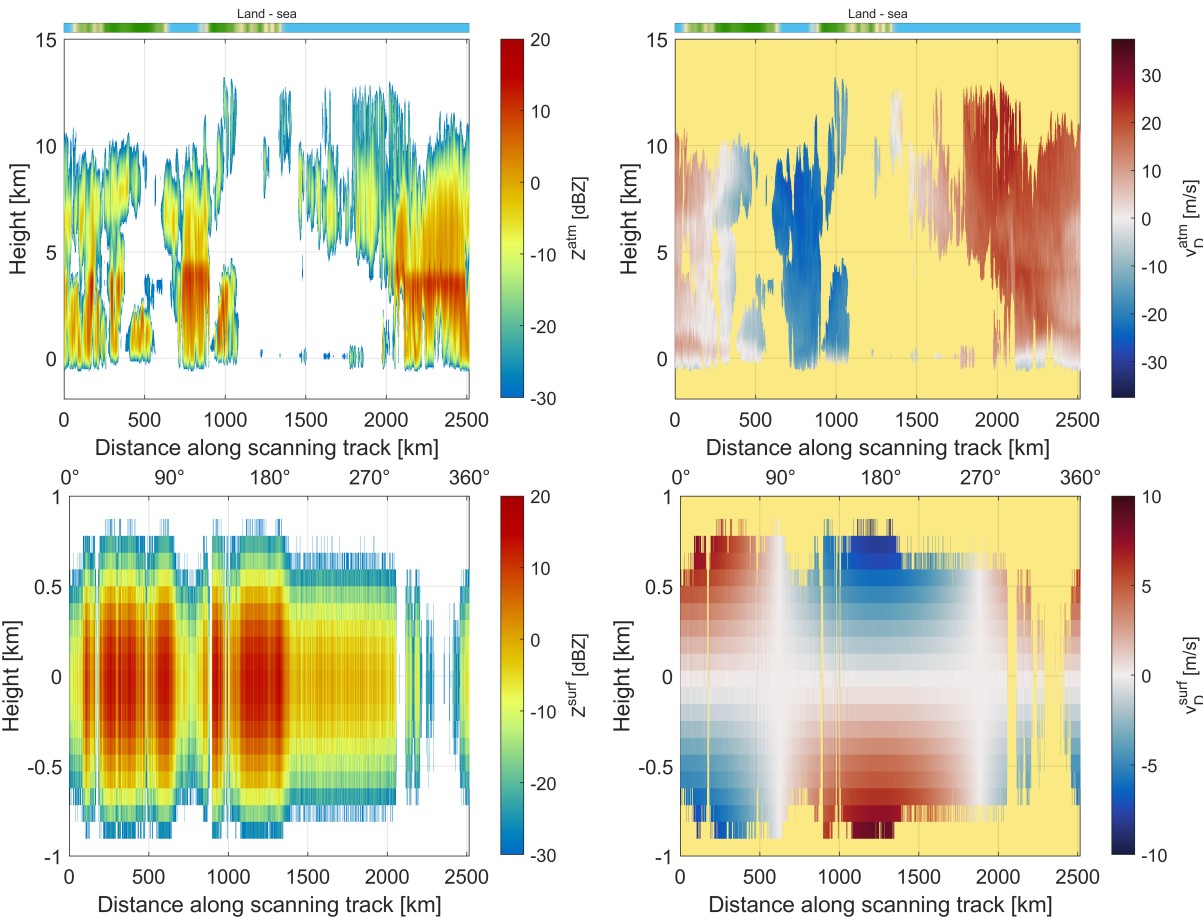

**Figure 13.** Atmospheric (top) and surface (bottom) reflectivities (left panels) and Doppler velocities (right panels) in correspondence to the revolution shown in Fig. 9B as a function of the distance along the scanning track. In order to facilitate the interpretation we have added on top of the top panels colorbars indicating surface type (green: land; blue: ocean; brown: coasts) and on top of the bottom panels labels indicating the azimuth position angle $\phi$ (measured clockwise and equal to zero when the antenna is looking forward along the travelling direction, see Fig.5). For the velocity panels we have used a yellow background in correspondence of low $SNR$ regions for clarity of display. The reflectivity (top left panel) clearly shows regions of high attenuation below the freezing level altitude (located at about 4.5 km) especially in correspondence to distances along scanning track between 2200 and 2300 km. In that region the attenuation is so high that the surface contribution is well below the radar sensitivity (bottom left panel). In the lower panels the surface contributions are shown in the $\pm 1$ km altitude region. Note that the ground clutter for the 500 m long pulse is higher over the land than over the sea and higher in regions with no attenuation (bottom left panel).

Fig. 14). The ghosts only marginally affect the LOS velocities (Fig. 15, right panel); they only cause an increase in the standard deviation of the Doppler velocities according to Eq. (16) in the regions with detectable signal. Velocities are limited to the Nyquist interval (-40, +40) ms$^{-1}$ which is broad enough to capture the maximum amplitude of the LOS winds in this scene

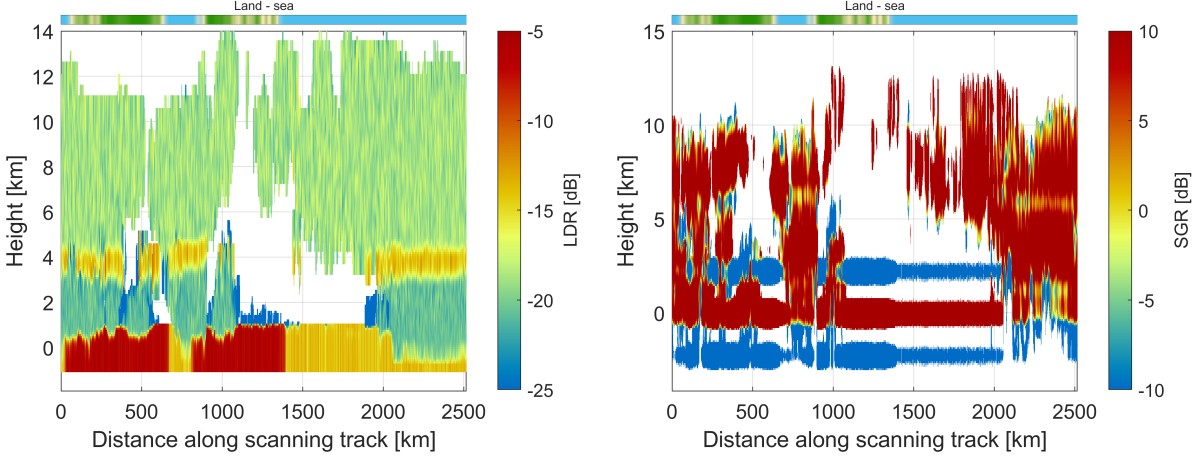

**Figure 14.** Linear depolarization ratio ($LDR$, left) and signal to ghost ($SGR$, right) in correspondence to the revolution shown in Fig. 9B as a function of height. The $LDR$ clearly shows the significant depolarizations by the melting layer that is straddling the heights around 4 km and by the surfaces with clear transitions from strong depolarizing land to weaker depolarizing sea surfaces. At ranges where ghost but no real clouds are present the $SGR$ becomes $-\infty$ in logarithmic units (zero in linear units); thus the $SGR$ is capped at -10 dB. This is the case for several instances at an height of $\pm 2.3$ (which corresponds to a slant range of 3 km) in coincidence with surface-cross talk.

(Fig. 12B). In regions with very low $SNR$ or $SGR$ (e.g. around -2.3 km below the surface) the estimated velocities practically become random numbers within the Nyquist interval (hence the grainy texture in the graph). Otherwise the estimated LOS Doppler velocities well resemble the LOS winds depicted in Fig. 12B, which is confirmed by the good precision of the wind,s always better than 3 ms$^{-1}$ in regions of high $SNR$, and by small biases introduced by NUBF and wind-shear errors (see

5    Sect. 3.2 later on).

   Another WIVERN product is the H and V-polarized brightness temperatures (Fig. 16). Due to the difference in emissivities there is a clear separation of the vertical and horizontal polarized $T_B$ over the ocean. With increasing optical thicknesses the two $T_B$ tend to get closer and closer. This $T_B$ enhancement due to emission over cold backgrounds is expected to be useful for rain retrievals. In fact, because of the reduced and more variable ocean NRCS, surface reference technique-based path integrated

10   attenuation (PIA, Meneghini et al. (2021)) estimates will be more challenging and more sparse in WIVERN configuration than for nadir-looking radars. In addition $T_B$ are known to have a better sensitivity than PIAs (Battaglia et al., 2020a), i.e. they will produce a detectable signal at smaller optical thicknesses (compare blue and red line variability). The coincident sampling of reflectivity profiles and $T_B$ will be unique and provide insights into supercooled cloud liquid water coexisting with snow over the ice-free ocean (Battaglia and Panegrossi, 2020) and the evolution of large ice particles in deep convection.

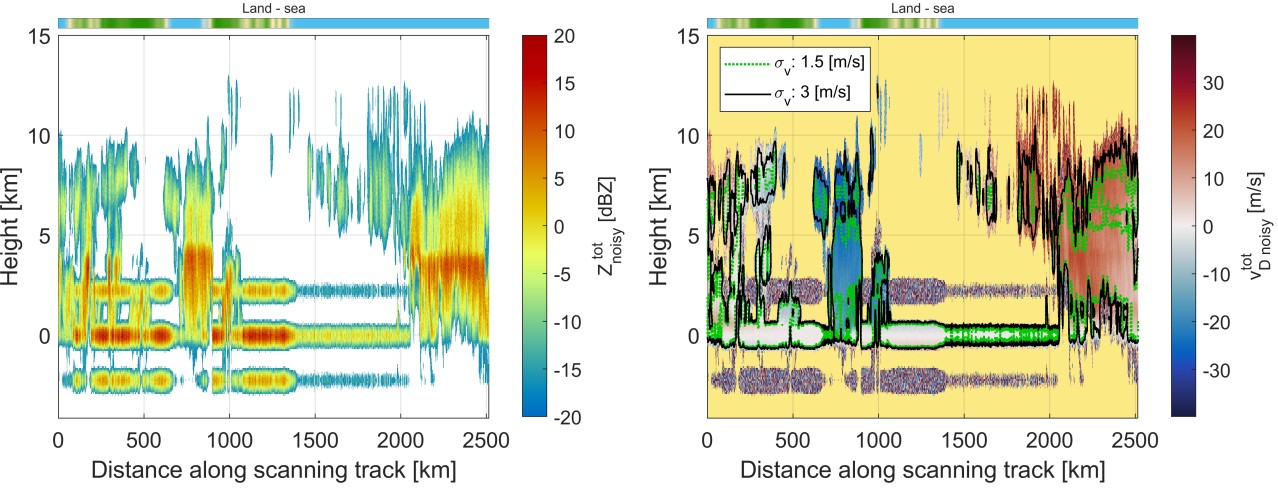

**Figure 15.** Reflectivities and Doppler velocities results corresponding to a full revolution of the WIVERN antenna as shown in Fig. 13 for an integration length of 1.0 km ($M$=8). Left panel: simulations of the WIVERN reflectivities (signal+noise) with ghost echoes at 2.3 km above and below the ground due to the depolarization by the surface leading to ghost echoes where there is no real cloud. Right panel: WIVERN retrieved line-of-sight Doppler velocities after performing the polarization diversity pulse pair processing. Black solid and green dashed contour lines correspond to Doppler velocity precision computed according to Eq. (16) of 3 and 1.5 ms$^{-1}$, respectively. Note that the echoes contaminated by ghosts have lower precision.

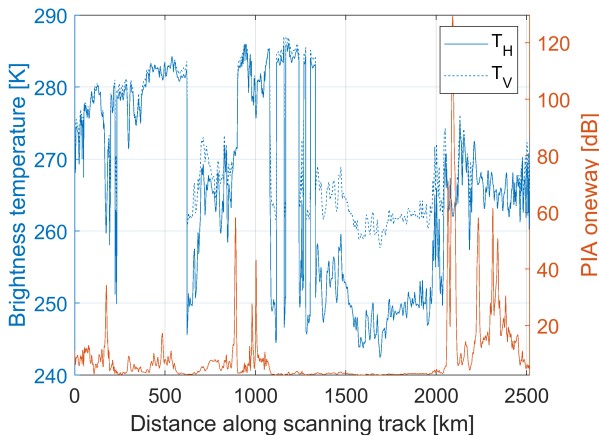

**Figure 16.** Simulated brightness temperatures for $V$ (continuous blue) and $H$ (dashed blue) polarization. For clarity of presentation we have not added the expected measurement noise. The total one-way PIA (red line) is also given for reference. Results correspond to a full revolution of the WIVERN antenna as shown in Fig. 13.

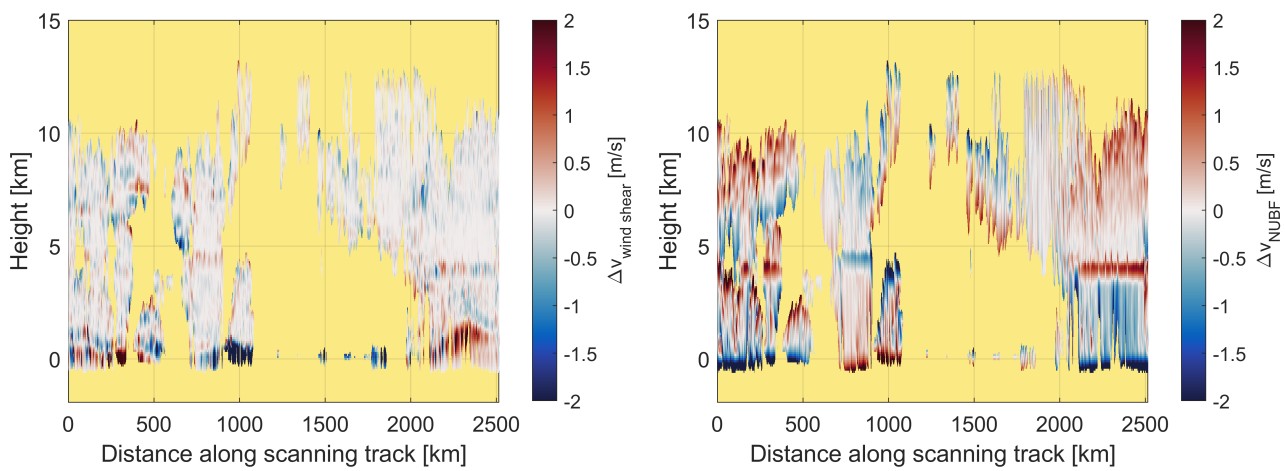

**Figure 17.** Errors induced by wind shear (left) and NUBF (right) in correspondence to the revolution shown in Fig. 9B.

### 3.2 WIVERN performance assessment

The E2E simulator represents a useful tool to study the performances of the WIVERN mission. Apart from the errors related to the Doppler estimators in the pulse-pair processing (Eq. 16) and the mispointing (Tanelli et al., 2005) there are other sources of uncertainties in polarization diversity Doppler radar measurements such as errors linked to wind shears either associated with the platform motion (Tanelli et al., 2002; Kollias et al., 2014) or to the atmospheric winds (Battaglia et al., 2018), to clutter contamination (Illingworth et al., 2020), to aliasing (Battaglia et al., 2013; Sy et al., 2014). The contribution of each of these errors can be quantified unambiguously by running two simulations where the effect is turned on and off.

#### 3.2.1 Wind shear errors

The wind shear errors which tend to occur when reflectivity and velocity gradients are present at the same time within the backscattering volume, as can happen at the boundaries of clouds, can be computed from the difference between $v_{AW}$ in Eq. (19) and the expression of $v_D^{atm}$ in Eq. (14) with $v_{sat}$ set to 0. Results are shown in the left panel of Fig. 17 in correspondence to the revolution shown in Fig. 9B. Strong wind shears appear in this case at near-surface altitudes (see Fig. 12b). This results in significant wind shear errors exceeding $\pm 1\,\mathrm{ms}^{-1}$ affecting the measurements at the low altitudes, but Fig. 17 shows that these errors impact only limited regions and are close to zero for most areas within the observed scene.

#### 3.2.2 Non-uniform beam filling: satellite motion-induced biases

Estimates of the NUBF errors can be obtained by comparing the expression of $v_D^{atm}$ in Eq. (14) with and without $v_{sat}$ set to 0. Because of the different directions of the satellite velocity with respect to the antenna boresight when changing the scanning position, the NUBF errors depend on the azimuth scanning angle. The satellite velocity produces an apparent wind shear across

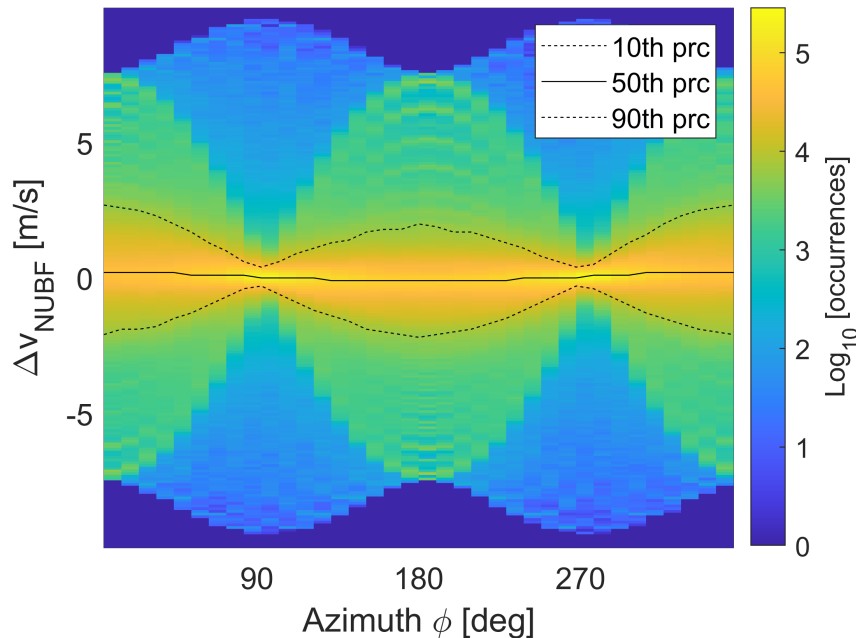

**Figure 18.** Histogram of NUBF-induced error as a function of the azimuthal scanning angle. The color indicates the $\log_{10}$ of the number of occurrences. The statistics is computed for 20 full rotations over the scene depicted in Figs. 12-15. The dotted lines represent the 10th and 90th percentile (typically with absolute value lower than 2.0 ms$^{-1}$) whereas the continuous line corresponds to the median value (always very close to zero because many NUBF errors are equal in magnitude and opposite in sign so will tend to cancel out).

the backscattering volume, with velocities ranging from -3.4 (-4.4) to +3.4 (4.4) ms$^{-1}$ at forward/backward- (side-) looking configurations across the 3 dB WIVERN footprint. When coupled with a reflectivity gradient (computed along the direction orthogonal to the boresight and lying in the plane generated by the satellite velocity and by the antenna boresight (Battaglia et al., 2013)) this satellite-motion-induced velocity shear will produce a NUBF bias. The right panel of Fig. 17 shows NUBF

5   errors for the revolution of Fig. 9B and clearly demonstrates that the effect can be of several ms$^{-1}$, is strongly azimuth angle dependent (typically minimised at side view, e.g. for a distance along scanning track of about 625 and 1250 km) and is driven by vertical gradients (e.g. strongly enhanced at the cloud top and in the melting layer). We have computed NUBF statistics for 20 full revolutions (i.e. a distance along scanning track exceeding 50,000 km) over the Labrador scene depicted in Fig. 9A: the distribution of the NUBF Doppler velocity biases as a function of the azimuthal scanning angle is shown in Fig. 18. In general

10  the effect is maximum in the forward and backward directions ($\phi = 0°$, $180°$) because in these directions the error is partly coupled with the vertical gradients of reflectivity. As a result, opposite biases are generally present in the upper troposphere and in the bright band (where $Z$ is sharply decreasing with height) and in the lower troposphere where, due to attenuation, reflectivities are increasing with height. At side view ($\phi = 90°$, $270°$) the error is coupled only with the horizontal gradients

of reflectivity, which may not be well captured by the model due to its coarse horizontal resolution. Thus NUBF errors may be underestimated by our simulations at side view. Overall NUBF errors are within the requirements with the 10th and 90th percentiles exceeding $2$ ms$^{-1}$ only at backward and forward viewing. Due to the conically scanning symmetry NUBF errors are equal and opposite when considering corresponding scanning directions in the forward and backward segment of the scan. When averaging winds over spatial scales of the order of 20 km or more (see Tab. 1) this will practically eliminate the NUBF bias and only worsen the precision of the LOS winds.

## 4   Conclusions and future work

This study introduces a state-of-the-art E2E simulator tailored to simulating space-borne conically scanning Doppler radars adopting polarization diversity with the inclusion of a radiometric mode. The "WIVERN" configuration as proposed to the ESA-Earth Explorer 11 call (see specifics in Tab. 3) has been implemented in this study. The simulator reproduces the satellite orbit, the radar scanning geometry and the illumination of an atmospheric scene extracted from a global atmospheric circulation model providing fine resolution vertical profiles of winds and clouds. The coupling between the orbit and the atmospheric model allows global scale simulations of mission observables, i.e. reflectivities and Doppler velocities of atmospheric targets. In addition, surface modelling accounts for the clutter returns from land and sea surfaces. The simulator also outputs estimates of Doppler measurement errors, such as those due to intrinsic noise, to cross-talk noise between the two diversely polarized channels and introduced in presence of reflectivity gradients (wind shear and non-uniform beam filling errors). Additional disturbances originate from the antenna azimuthal mispointing errors, represented in terms of an absolute knowledge error power spectrum.

Preliminary findings show that mispointing errors associated with the antenna azimuthal mispointing are expected to be lower than $0.3$ ms$^{-1}$ (and strongly dependent on the antenna azimuthal scanning angle), wind shear and non-uniform beam filling errors have generally negligible biases when full antenna revolutions are considered, NUBF causes random errors strongly dependent on the antenna azimuthal scanning angle but typically lower than $1$ ms$^{-1}$ and cross-talk effects are well predictable so that areas affected by strong cross-talk noise can be flagged. The noise random errors are dependent on the $SNR$ and the possible presence of ghosts and can be reduced by averaging over a higher number of pulses (i.e. by using a longer integration time). In summary our results show that the quality of the Doppler velocities appears to strongly depend on several factors: the strength of the cloud reflectivity, the antenna pointing direction relative to the satellite motion, the presence of strong reflectivity and/or wind gradients, the strength of the surface clutter. Overall, the E2E simulations suggest that total wind errors meet the mission requirements in a good portion of the clouds detected by the WIVERN radar, which is a very encouraging finding at the beginning of Phase 0 studies.

The characterization of the errors and the isolation of each single error source makes the E2E simulator a useful tool to verify mission performances and compliance with requirements, which will be part of the Phase 0 studies started in December 2021 and due to an end in October 2023. Different problematic areas will be investigated with the introduction of new features (see Tab. 2).

1. By changing the antenna gain (Eq. 1) it will be possible to study the impact of antenna side-lobes in affecting the minimum height close to the surface at which winds can be observed by the WIVERN radar without suffering significant biases from the clutter return.

2. More sophisticated surface modelling could be introduced by including the dependence on the surface winds over ocean and different surface types over land.

3. Cloud scenes at finer horizontal resolution ($\lesssim 1$ km) that resolve convection could be used in the simulator at regional (if not global) scale; this will enable to evaluate WIVERN performances in presence of convective motions.

4. A multiple scattering module based on the two-stream approximation (Hogan and Battaglia, 2008) could be applied to the 1D WIVERN slant column and used to flag multiple scattering contaminated profiles in regions of deep convection.

5. Additional polarimetric variables like specific differential phase ($K_{DP}$), specific differential attenuation ($A_{DP}$) and cross correlation ($\rho_{hv}$) could be simulated. This requires fundamental changes in the scattering LUTs and in introducing polarization dependency in all variables.

6. Further studies on mispointing effects will be performed once power spectral densities of azimuth and elevation knowledge error will be better specified by industrial studies. In particular the E2E simulator will be able to assess how frequently and with which accuracy the surface return could be used to check the elevation pointing.

7. The E2E simulator will also serve as the basis to develop mitigation algorithms for NUBF, wind shear, mispointing and vertical wind corrections that will be needed in order to produce horizontal line of sight winds, which will be the product directly assimilated by numerical weather prediction models.

Thanks to its modular structure the simulator can be easily adapted to different orbits, a gamut of scanning geometries (e.g. cross track) and various frequencies (by simply changing the look-up-tables). Therefore the simulator could be applied to simulate other space-borne Doppler atmospheric radars as well.

*Acknowledgements.* This work was supported in part by the European Space Agency under the activity Doppler Wind Radar Science Performance Study, ESA Contract no. 4000130864/20/NL/CT. PM's work was funded by Compagnia di San Paolo. This research used the Mafalda cluster at Politecnico di Torino.

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

*Data availability.*   The simulation inputs are available on request. The E2E simulator code is not yet available because part of on-going ESA studies.

*Author contributions.*   AB wrote most of the text and has built most of the modules of the simulator. PM has implemented the code in matlab and produced most of the figures. EC and LP have provided inputs for the orbital model and to the mispointing model plus they contributed to the scientific discussion. FS has provided supervision to PM and participated to the discussion on radar mispointing issues. PK and AI contributed to the discussion, the editing and the formulation of the WIVERN idea and definition.

*Competing interests.*   The authors declare that they have no conflict of interest.