# Peer review of "Observation error analysis for the WIVERN W-band Doppler conically scanning spaceborne radar via end-to-end simulations"

_Atmospheric Measurement Techniques, 2021_

## Author Response (AR1)

We thank the reviewer for the insightful comments.

Here after you will find the reviewer comments in bold, our replies in italic.

**One thing that could aid understanding could be a brief description of different types of radar models. As I understand it, the model described here simulates mean quantities and uncertainties, followed by generation of noisy samples (see my comment below). Perhaps a discussion of why this is used versus simulation of, for example, received amplitudes from collections of point scatterers. Another possible discussion would be validation of the model. Besides the one example in Section 3, were there other tests, either similar to Section 3 or perhaps idealized cases?**

*Yes indeed the model simulates mean quantities and errors as computed from theory for polarization diversity pulse pair. These errors proposed by Pazmany et al, 1999 have been confirmed by an airborne field campaign (Wolde et al, 2019). There are other simulators that compute I and Q time series like in Battaglia et al., 2013. This solution is quite demanding in terms of computational costs. There are no other simulators for conically scanning polarization diversity W-band spaceborne radars to our knowledge. We have tested our model with simple 1D scenes but a thorough validation is not that straightforward.*

**Are there plans to go beyond Mie theory?**

*Indeed the code accept look-up-tables (LUTs). Some of our LUTs are already computed with Rayleigh-Gans approximation, so there is no hurdles to include e.g. DDA computations. For preferentially oriented hydrometeors and dichroic media things are more complicated and they require a polarization-dependent treatment of backscattering and extinction. Some text to clarify this has been introduced in Sect.2.2.2. This is planned as future development (see new Table 2).*

**Antenna pattern – for example, a sinc function would be approximated as a Gaussian main beam plus sidelobes?**

*Yes, we have now implemented an antenna pattern with sidelobes. An example of such antenna has been included in the revised version. No appreciable differences have been noticed with such low level of sidelobes.*

**On page 12, line 20 "convoluted" is maybe better "convolved".**

*Corrected*

**My understanding from pp. 15-16 is that the theoretical uncertainties are used to generate properly distributed noise that is combined with the calculated means. Is this correct? Does the code also output the underlying means and uncertainties?**

*Yes exact, we have rewritten part of the text to make this more clear. In Fig. 15 right panel you can actually see the standard deviation errors associated to the Doppler velocities.*

**Maybe more details could be provided on the simulated brightness temperature, such as bandwidth, integration time, and resulting uncertainty. Are the brightness temperature samples computed the same way as radar observations, namely, by generating means and then adding noise?**

*Yes correct. Bandwidth and integration time are relevant for the computation of uncertainty. Uncertainties of the order of 3 K are expected for 5km integration and the current of bandwidth. .*

**I initially got confused by the text at the bottom of page 18, which mentions "Panel B" and Figures 11A and 11B. As stated it's all correct, but, for clarity, perhaps the discussion of Figure 8 could be its own sentence. This could be followed by new sentence, such as "For this full scan circle, Figure 11 shows the antenna weighted hydrometeor water content, as computed using the following".**

*Amended.*

**In Figure 12, why is the surface Doppler (height 0) in the lower right panel so weakly modulated by azimuth angle?**

*There is no azimuthal dependence expected for the Doppler velocity of the surface, which is assumed to have zero velocity. Note that the satellite velocity along the boresight direction is always subtracted out.*

**P. 21, around line 12 – not sure I understand the comment that "the ghosts are significantly smaller over land than over ocean". The effects in Figure 14 seem larger over land.**

*Thanks for pointing it out. Of course the sentence should read: that "the ghosts are significantly smaller over ocean than over land".*

**p. 25, line 10 – "20 full revolutions" – is this the same mean reflectivity and velocity but different noise or this is 20 different scenes from the full track?**

*20 different turns. We have clarified this in the text.*

We thank the reviewer for the insightful and detailed comments.

Here after you will find the reviewer comments in bold and our replies in italic.

**According to title and abstract, the paper aims to present an end-to-end simulator for the scanning Doppler radar WIVERN proposed to the ESA Earth Explorer program. Particularly with this aim, I find the paper too unspecific and lacking details, ie failing the traceability criteria. Also, the differences and novelty compared to Battaglia et al. (2018) do not become sufficiently clear to me.**

*We have tried to be more specific and detailed in the revised version. The main difference from the Battaglia et al paper is that in that case the simulation used a very simple 1D scene (reconstructed from CloudSat). In the current implementation the full 3D scanning geometry of the WIVERN satellite is implemented. This allows to account and assess how the quality of the Doppler signal will depend on the azimuth viewing angle. Also the simulator is now coupled with a full 3D cloud output which allows to evaluate the importance of non uniform beam filling errors. Finally, there is an orbit model coupled to the radar instrument model. This for instance allow to link the instrument model to errors like the mispointing errors which are orbit related. All these are completely novel aspects. See new text at bottom of page 3 and top of page 4.*

**On the other hand, I find the paper to focus a lot on the WIVERN instrument and its observation error analysis. While this is very valueable and fits the scope of AMT (while a simulator-focused paper would better fit into GMD according to my understanding), it should be reflected more clearly in title and abstract.**

*Comment received. It is true that we are focusing more at the observation error analysis. We have changed the title to "Observation error analysis for the WIVERN W-band Doppler conically scanning spaceborne radar via end to end simulations") and the abstract accordingly.*

**Major comments:**

**The introduction elaborates on aims and novelties of the WIVERN mission (far too much in my opinion, since this is supposed to be a simulator-paper, not a WIVERN mission paper), however I miss putting it in context with the past and current sat-borne radar missions CloudSat and, due to its Doppler capabilities in particular, EarthCare. Also, it lacks a definition of what is meant by "end-to-end simulator", incl. what it distinguishes from satellite, observation, or forward simulators or operators (at least in the understanding and usage of the authors) and a review of the state of the art in such simulators or operators. In that context, a definition or explanation what the authors mean by "polarization diversity" could be helpful, too.**

*We have introduced a paragraph where we explain the key differences with other simulators currently available (end of page 3 beginning of page 4). The Earth CARE simulator in particular is for a pulse-pair (not for a polarization diversity pulse pair system) Doppler radar, and it is not for a conically scanning. In its current version it has no mispointing error characterization, and a very crude surface modelling.*

**From the intro of Sec2, it is unclear to me whether the referenced literature describes approaches in general, or a specific algorithm or implementation of a module, and the following subsections do not make it clearer. Also, please distinguish between options available in the E2E simulator and specific setups used here.**

*These are the different modules implemented in the E2E simulator. Some of them are based on previous work and described in the literature. We have rewritten some of the introductory part of section 2.*

**It remains unclear, what the exact requirements are on the model input incl. which parameters are needed (which hydrometeor parameters specifically? temperature? etc.). Are the SAM data described in subsec 2.1 the only data the E2E simulator is/can be used as model input, or is this "just" what is used in the application examples later on?**

*SAM is what is used at the moment. We are currently working at interfacing the code with WRF. In principle any geolocated model output can be ingested. The model output needed are temperature, pressure and relative humidity plus the different hydrometeor contents (and particle size distribution assumption). This has been now specified in*

**Subsec 2.2.1 details the planned WIVERN orbit and observation geometry. However, how is this implemented in the E2E simulator? Are, e.g., the orbits hardcoded or can orbit parameters be changed, ie different orbital setups be explored? If so, what can the user specify?**

*Yes the orbit parameters can be changed. The simulator implements an orbital model deriving from the two-body problem theory, with the addition of orbital perturbations due to the $J_2$ effect to simulate Sun-Synchronous orbits. The user can indeed modify the initial date and duration of orbital propagation and the orbital parameters. Knowing the satellite position vector over time and the scanning method, a vector-based approach is followed to localize the antenna boresight direction and the illuminated region of the atmosphere and the surface.*

**Subsec 2.2.2 lacks almost all useful details about the scattering lookup tables like: which parameters are tabulated, bulk or single scattering properties? Over which tabulation parameters? where does the size distribution information come from and how is it taken into account? what dielectric property assumptions are made? how can lookup tables be generated, e.g. to switch to other scattering approximations like the mentioned Rayleigh-Gans?**

We have bulk extinction, backscattering and scattering coefficients tabulated per unit mass as a function of characteristic size (mean mass-weighted diameter) and mu (Gamma functions are used). The model also use exponential functions for PSDs with specific assumptions on N_0. We have included all the requested details.

**How are the empirically derived LDR linked to Mie reflectivities, is there anything to ensure a certain level of consistency? As LDR are derived based on ground-based observations - are they comparable to sat-borne measured ones?**

*AT the moment the LDR is not consistent with reflectivities but it is simply based on climatological observations. The LDRs are relevant when considering the cross-talk effects; at this stage we only want to see what is the climatological impact of the ghosts, thus we believe this is enough at this stage. We do not have any polarimetric observation from space-borne radars (even at lower frequencies) but there is no reason (apart from the increased footprint size and increased levels of multiple scattering) why sat-borne measurements should be different from ground-based ones. A new sentence has been introduced at the end of Sect.2.2.2.*

**Table 4 is never discussed nor mentioned in text. It's completely unclear what it is presenting and why it is there.**

*Yes Table 4 has been deleted. Thanks for spotting it.*

**Figure 6 seem to indicate that a plane parallel atmosphere model is used - is that so? Also, is the beam lobe modelled with a constant solid angle or a geometric distance opening (given values in meters, the figure seems to indicate the latter).**

*No, the model is full 3D, so the antenna pattern is currently modelled as a Gaussian main lobe. The simple plane parallel atmosphere was implemented in the 2018 version. The caption of figure 6 has been modified*

**For subsec 2.4, please give a short explanation what pulse pair processing is (or, what you mean by that).**

*The Doppler velocity in radar systems is derived by measuring phase shifts between successive pulses (pairs). We have introduced some explanation and references. See new text in Sect.2.3*

**Does the E2E simulator for the radiometric mode shortly mentioned as subsec 2.6 consider gas absorption/emission, too, or just hydrometeors scattering and emission/absorption contributions? If the first, what absorption model is used?**

Yes gas absorption is included. The Rosenkraz model is used. We have included appropriate references to it.

**For the case study (subsec 3.1), please be more specific: what date and time is that? what is the general weather situation? Where is the reader supposed to see "some strong wind shear" in the modelled scene?**

You can see the strong wind shear in Fig.12 top right panel in the bottom right section of the scene.

**For the figures in general, please consider the use of color schemes that are suitable for people with color vision deficiencies, preferably such that provide perceptual uniformity.**

*All figures with colorbar (12,13,14,15,17) have been changed. We have used colorbars recommended by https://ntrs.nasa.gov/api/citations/20180004634/downloads/20180004634.pdf.*

**For the case study figures, to allow easier comparison, please be so kind to use the same x-axis (incl. same axis parameters and units) for all of them (if using azimuth, axis ticks & labels at 90° spacing would be nicer and support interpretation better). Moreover, when discussing specific patterns in a figure, refer to the axis parameter used in that figure (in text, surface reflectivity is referred to in along track coordinates, while the plot is in azimuth coordinates).**

*Ok we have amended the figures accordingly (see new Fig.12)*

**For the list of problems to investigate in the future in Sec4, that by the way is quite specific compared to the rather indistinct description of the current state of the E2E in Sec2, it would be interesting to know, which problems require additional simulator development/implementations and which are rather setup changes.**

*We have added some text to explain this in Sect. 4.*

*Thanks for the list of specific comments that have been implemented in the revised version.*

---

## Author Response (AR2)

We thank the reviewer for the insightful comments.

Hereafter the reviewer's comment in bold and our reply in italic.

**p. 4, around line 15: I think lines 11-15 are one sentence. The authors could consider splitting into two.**

*We have split the sentence.*

**Table 2 lists "Gaussian" as the current pattern, but from Fig. 7, right panel, it seems that the model can already handle more complicated patterns. Fig. 7 also suggests that point-target responses besides top-hat can currently be used.**

*Yes indeed the E2E has been updated to do that, so we have taken out from the Tables thw two items mentioned by the reviewer.*

**p.9, line 5 – the start of a new paragraph here isn't really needed; it looks like it might be due to the edit at the end of the sentence ending with "to Rosenkranz (1998) model". Also, probably better "to the Rosenkranz (1998) model". In general, I noticed some very short paragraphs that could be combined with surrounding paragraphs.**

*This has been done.*

**The numerical weather model probably puts out the mean atmospheric quantities at each grid point. The bottom of p. 6 mentions "particle size distribution assumptions". This is further mentioned at the top of p. 10. However, the steps in taking model output to particle size distribution could be clearer.**

*We have rephrased some of the sentences*

**p. 18, line 17: "must be performed before the deconvolution in range (Eq. 5)". Should that read "convolution" rather than "deconvolution"?**

*yes, thanks for spotting this*

**p.22 bottom and p. 23 top: subtraction of satellite motion is mentioned twice. Maybe these can be combined into a single mention of this.**

*One has been deleted.*

We thank once again the reviewer for the insightful and detailed comments he/she has provided.

Hereafter the reviewer's comment in bold and our reply in italic.

**Furthermore, I think the outlook part of the Conclusions section should be adapted to standards of a scientific paper (this is not a proposal) and be made significantly more concise: it's out of proportion compared to the rest of the section (makes up about 50%), it is not reflected sufficiently in the main part of the paper (eg in the form of discussed shortcomings/limitations of the current E2E and analysis method), and it is mostly a wish list (where it remains unclear how much implementation work is needed, what is the level of intention of realisation, on what time frame that is supposed to be realised and how much that might affect the current analysis) that is of no practical use for the reader. I still find the distinction blurry between what is implemented in the E2E on the one hand and what is setup for this study on the other hand. Table2, e.g., mixes them up in my understanding (the caption states "simulator capabilities" ie implementation, but later figures suggest other than top hat PTR are available, ie top hat PTR is the setup chosen for this study).**

*The future work section has been shortened. Table 2 also has been modified and some of the features eliminated. Discussion about the remaining issues (multiple scattering, polarization) has been added in the text.*

**Throughout the manuscript, please**
**- use consistent typesetting of parameter symbols (e.g. LDR is at times set in normal font, most times in italics)**
**- typeset units in normal font, not italics**
**- spell out all abbreviations, acronyms, and parameter symbols at first(!) occurrence (this includes but is not limited to NUBF, PIA, SST, SGR, I/O)**
**- check that appropriate (in-text or in-parentheses) citation styles are used**
**- replace any appearances of "the Doppler" with an appropriate phrase (velocity? spectra? ...?)**
**- replace the use of semicolon as punctuation mark unless it really serves a purpose. It can usually be replaced by a period and starting a new sentence (long sentences with several subclauses make it hard for the reader to understand)**
**- refrain from using brackets in text in place of parentheses**
**- reduce the use of entire text clauses in parentheses. This makes the text hard to comprehend.**
*We have followed all reviewer's suggestions*

**Add references for the WIVERN mission provision expectations (P2L21ff) and the WIVERN science objectives (P2L29).**
*Done*
**A short definition of end-to-end simulator would be helpful (P4L8).**
*Short sentence has been introduced.*
**P1L17ff: This seems to be a leftover from the initial draft. It's redundant with E2E summary in P1L5ff.**

*The sentence has been shortened.*

**P1L15: "The total wind errors seem to meet" - Reformulate in a more scientific manner ("seem" seems**

inapproriate for a scietific paper), e.g. "The end-to-end simulations suggest that the errors..." (same re-appears in P28L12)

*Reformulated*

**P2L13: "a Dual-polarization" -> "a dual-polarization**

**P2L14: "a 3-m cirular" -> "a 3m circular"**

**P3L19f: "non uniform" -> "non-uniform"**

**P4L4: "Battaglia et al. (2018) also used" -> Remove "also" to avoid confusion (also as in "in this study, too" or as in "furthermore"?).**

**P4L31: "Sun-synchronous" -> "sun-synchronous"**

*All done*

**P5L8: "A global model" - is really a global model needed, data from mesoscale or local-area models do not suffice?**

*Well is not needed but it is certainly good to have a model that can cover all latitudes and longitudes covered by the satellite.*

**P6L1: "These estimates have been been confirmed" - validated?**

**P6L1: "by airborne" -> "an airborne"**

**P7 Tab2:**

**- Why is model resolution a "capabilitiy" of the E2E - can't it handle any other (or higher?) resolutions than 4.3km at the moment (due to memory and/or computation time requirements?)?**

**- sigma_0 not yet explained nor mentioned(better to be spelled out here?); similarly the simulated radar variables (spell out in caption?).**

*All corrected.*

**- What is the multiple scattering flag? Multiple scattering is not discussed anywhere else in the manuscript (apart from the outlook), ie its role and relevance remains completely unclear. I suggest to remove it, apart maybe a mention as current limitation of the method in the conclusion section. (similar applies to ZDR, ADP, KDP - completely unclear what their roles and benefits would be).**

*We have introduced two additional sentences to explain the importance of multiple scattering and polarimetry.*

**P8L3: "output from the GSRM" - "from a", maybe? Or did only one GSRM participate in DYAMOND (the previous paragraph suggests differently)?**

**P9L3: Spell out gamma.**

**P10L2ff: "with different densities and axial ratio" - This whole part is too unspecific to be of use for the reader. I suggest to focus here on what has been used in the study (Mie spheres), and to mention non-sphericity and polarimetry at max in a discussion of limitations (hence desireable refinements) of the methods (if they are expected to play a role).**

**P10L11f: "in the area of temperatures" -> in the assumed melting layer (at temperatures ...)"**

**P10L15: "sufficient to see" -> to demonstrate? analyse?**

**P12L2: "The PTR could be assumed" -> is assumed?**

**P13 Fig7 caption: "different PTR", "different antenna patterns" -> remove "different" (showing the same would make no sense, would it?)**

**P13L13: "H and V pulse" -> pulses; hence subsequent verbs in plural conjugation**

**P14L2: "interested to" -> interested in**

**P15L11: Provide a reference for Kw^2.**

**P1616: "The hatched regions" - Ref to figure that is seemingly discussed here is missing.**

*All done*

**P1625: What are the "M-pairs"? Not mentioned/explained before.**

**P16L27: "for the first M pulses of the pairs" - unclear what is meant. a pair is only two, ie has a first and a second pulse. how can anything be done with first M pulses? Does it mean the first pulse of each of M pairs, maybe?**

*We have amended this part referring to Fig8.*

**P17L2: "exceed 3 m/s" -> where does this number come from?**

*A formula has been introduced to explain it.*

**P17L6: "using the fact the" -> "using the fact that the"**

**P17L16: "is projection of the the satellite" -> "is the projection of the satellite"**

**P17L17f: "and Zco is the measured co-polar reflectivity factor" - in my understanding it is either measured (but than the sum of co- and cross-polar) or co-polar.**

**P18 Fig9: What is shown as backgound of Panel B (what is PIA?)?**

**P20L8: "Preliminary assessment suggests" - please add a reference.**

**P21L5: "The LOS winds [...] present some strong vertical wind shears" - check sentence structure; doesn't make sense.**

**Figs12ff: Clearly state once (in the text body, preferably) what is the meaning of "distance along the scanning track" (the first interpretation it triggers for me as along the satellite track, which however doesn't make sense; after some thinking I conclude it is the length along the ground projection of the rotating scans and the 2500km are equivalent to a 360° roatation)**

**P21L9: "are plotted" -> are shown**

**P21L11: for the strong attenuation examples, isn't the event at >2000km much more significant hence worth mentioning than at 1100km?**

**P21L15: "(see Fig...)" -> use in-paranetheses (bibtex' citep) citation style**

**P21L17 and P22L2: "the satellite velocity...substracted out" - redundant**

**P22 Fig13: Thanks for adding the land-sea and the scan azimuth info to the plot. Very helpful!**

*All done*

**P22 Fig13 caption: "Measured clockwise" - how is that consistent with that WIVERN scans counterclockwise (P20L13)?**

*It does not really matter what convention has been used for the azimuthal angle as far a swe understand where the antenna is pointing at.*

**P22 Fig13 caption: "In such region" -> "In that region"**

**P22 Fig13 caption: "so high to bring...well below" -> "so high that the surface contribution is well below"**

**P22 Fig13 caption: "and in regions with no" -> "and higher in regions with no"**

**P23L14: "regions with signal" -> signals**

**P23L15ff: "because of the considerably" - hard to read and as is does not make sense. reformulate. try to reduce the amount of comments in partentheses.**

**P24 Fig15 (and later figures): What is the meaning of the yellow background?**

**P24 Fig15 caption: "Continuous (dashed) black contour lines" - My version has black solid and green dashed lines. Correct, please.**

*All corrected.*

P24L6: "Otherwise the estimated LOS Doppler velocities well resemble the LOS winds depicted in Fig. 12B." - In my eyes the remblance is not particularly striking, hence this statement needs more (descriptive) support (or maybe a variant of 12B with reflectivity-weighted or -filtered LOS winds could help).

*Added a statement to explain*

**P24L8:** $T_B$ is a math symbol and in my understanding can not (or does not need to) be pluralised, ie "$T_{B}$s" -> $T_{B}$ (applies also to all further occurences of "$T_{B}$s").

**P24L14:** "supercooled liquid clouds in snow" - what is that? maybe "supercooled cloud liquid water coexisting with snow" or something alike?

**P25L3:** "[formula (16)]" - use proper (journal style conforming) reference to equation

**P26L4:** "For instance the wind shear" -> "The wind shear"

**P26L8:** "which results" -> result

**P26 Fig18 caption:** "NUBF-induced error" -> "Histogram of NUBF induced errors"

**P26 Fig18 caption:** "NUBF errors are equal and opposite" - doesn't make sense (equal and opposite are contradict each other). Reformulate.

**P27L2:** "Similarly, estimates" -> "Estimates"

**P27L13:** "along track distance exceeding 50,000 km)" - along which track? obviously not the satellite ground-track which comes first into (my) mind.

*All corrected*

**P28L23: Still unclear what Thv is. Not mentioned anywhere before. Is it relevant at all?**

*THV is a very important parameter because it defines the separation between the H and V-pulse. It is a key parameter in polarization diversity Doppler techniques. This is explained at top of page 15*

---

## Author Response (AR3)

**REPLY to EDITOR**

We have done the three minor corrections as requested.